# IRAK1-dependent Regnase-1-14-3-3 complex formation controls Regnase-1-mediated mRNA decay

Kotaro Akaki[1,2], Kosuke Ogata[3], Yuhei Yamauchi[4], Noriki Iwai[1], Ka Man Tse[1], Fabian Hia[1], Atsushi Mochizuki[4], Yasushi Ishihama[3], Takashi Mino[1], Osamu Takeuchi[1]*

[1]Department of Medical Chemistry, Graduate School of Medicine, Kyoto University, Kyoto, Japan; [2]Graduate School of Biostudies, Kyoto University, Kyoto, Japan; [3]Department of Molecular and Cellular BioAnalysis, Graduate School of Pharmaceutical Sciences, Kyoto University, Kyoto, Japan; [4]Laboratory of Mathematical Biology, Institute for Frontier Life and Medical Sciences, Kyoto University, Kyoto, Japan

**Abstract** Regnase-1 is an endoribonuclease crucial for controlling inflammation by degrading mRNAs encoding cytokines and inflammatory mediators in mammals. However, it is unclear how Regnase-1-mediated mRNA decay is controlled in interleukin (IL)-1β- or Toll-like receptor (TLR) ligand-stimulated cells. Here, by analyzing the Regnase-1 interactome, we found that IL-1β or TLR stimulus dynamically induced the formation of Regnase-1-β-transducin repeat-containing protein (βTRCP) complex. Importantly, we also uncovered a novel interaction between Regnase-1 and 14-3-3 in both mouse and human cells. In IL-1R/TLR-stimulated cells, the Regnase-1-14-3-3 interaction is mediated by IRAK1 through a previously uncharacterized C-terminal structural domain. Phosphorylation of Regnase-1 at S494 and S513 is critical for Regnase-1-14-3-3 interaction, while a different set of phosphorylation sites of Regnase-1 is known to be required for the recognition by βTRCP and proteasome-mediated degradation. We found that Regnase-1-14-3-3 and Regnase-1-β TRCP interactions are not sequential events. Rather, 14-3-3 protects Regnase-1 from βTRCP-mediated degradation. On the other hand, 14-3-3 abolishes Regnase-1-mediated mRNA decay by inhibiting Regnase-1-mRNA association. In addition, nuclear-cytoplasmic shuttling of Regnase-1 is abrogated by 14-3-3 interaction. Taken together, the results suggest that a novel inflammation-induced interaction of 14-3-3 with Regnase-1 stabilizes inflammatory mRNAs by sequestering Regnase-1 in the cytoplasm to prevent mRNA recognition.

*For correspondence:
otake@mfour.med.kyoto-u.ac.jp

Competing interests: The authors declare that no competing interests exist.

## Introduction

The expression of proinflammatory cytokines is the hallmark of innate immune responses against microbial infection. Whereas inflammatory responses are critical for the elimination of invading pathogens, excess and chronic inflammation can culminate in tissue destruction and autoimmune diseases. When innate immune cells encounter pathogen-associated molecular patterns (PAMPs) or damage-associated molecular patterns (DAMPs), they are sensed by pattern-recognition receptors such as Toll-like receptors (TLRs), triggering the transcription of inflammatory genes (*Fitzgerald and Kagan, 2020*; *Takeuchi and Akira, 2010*).

The expression of inflammatory genes is also controlled by post-transcriptional mechanisms to facilitate or limit inflammatory responses (*Anderson, 2010*; *Carpenter et al., 2014*; *Turner and Díaz-Muñoz, 2018*). Regnase-1 (also referred to as Mcpip1, Gene name: *Zc3h12a*), an RNase, is a critical regulator of inflammation. Regnase-1 binds to and degrades inflammatory mRNAs such as *Il6*

or *Il12b* by recognizing stem-loop structures present in the 3' untranslated regions (*Matsushita et al., 2009*; *Mino et al., 2015*). *Zc3h12a*-deficient mice exhibit an autoimmune pheno-type, indicating its importance as a negative regulator of inflammation (*Matsushita et al., 2009*; *Uehata et al., 2013*). Regnase-1 efficiently suppresses the expression of its target genes by degrad-ing CBP80-bound mRNAs during the pioneer-round of translation by associating with ribosome and a helicase protein, UPF1 (*Mino et al., 2015*; *Mino et al., 2019*). CBP80 binds to newly synthesized mRNAs in the nucleus and is replaced by eIF4E after the pioneer round of translation following mRNA export from the nucleus (*Maquat et al., 2010*; *Müller-McNicoll and Neugebauer, 2013*). Thus, it is possible that Regnase-1 recognizes target mRNAs in the steps leading to the pioneer round of translation.

The stability of cytokine mRNAs is dynamically regulated in innate immune cells under inflamma-tory conditions (*Carpenter et al., 2014*; *Hao and Baltimore, 2009*; *Turner and Díaz-Muñoz, 2018*). Post-translational control of Regnase-1 in response to inflammatory stimuli contributes to extending half-lives of inflammatory mRNAs. Stimulation of cells with TLR-ligands, IL-1β, or IL-17 results in the activation of IκB kinases (IKKs), which phosphorylate Regnase-1 at S435 and S439, in addition to IκBα (*Iwasaki et al., 2011*; *Kakiuchi et al., 2020*; *Nanki et al., 2020*; *Tanaka et al., 2019*). Reg-nase-1, phosphorylated at S435 and S439 is subsequently recognized by βTRCP, one of the compo-nents of the SKP1-CUL1-F-box (SCF) complex, which induces K48-linked polyubiquitination of Regnase-1, followed by proteasome-mediated degradation (*Iwasaki et al., 2011*). On the other hand, these stimuli also induce transcription of *Zc3h12a* (*Iwasaki et al., 2011*). Consequently, the protein level of Regnase-1 drastically changes during these stimulations; Regnase-1 levels decrease immediately after the stimulation and then increase to levels higher than its pre-stimulation. How-ever, the post-translational regulatory mechanism of Regnase-1 following inflammatory stimuli is still not fully elucidated.

14-3-3 family proteins are conserved among species and are known to form hetero- or homo-dimer (*Aitken, 2006*; *Pennington et al., 2018*). The 14-3-3 dimer binds to various phosphorylated proteins using its two phosphor-S/T binding pockets which recognize unique phospho-peptides (*Muslin et al., 1996*; *Yaffe et al., 1997*). Although 14-3-3 itself has no enzymatic activity, 14-3-3 is known to modulate the properties of target proteins, such as protein stability or localization (*Aitken, 2006*; *Pennington et al., 2018*).

In this study, we utilized an interactome-based approach to isolate Regnase-1 protein complexes and found that TLR-ligand, IL-1β, or IL-17 stimulation induces the formation of the Regnase-1-14-3-3 complex. The phosphorylation of Regnase-1 at S494 and S513 is responsible for binding with 14-3-3, which in turn stabilizes Regnase-1 protein by excluding βTRCP. However, 14-3-3-bound Regnase-1 is not functional because 14-3-3 prevents Regnase-1 from recognizing target mRNAs. In addition, we found that nuclear-cytoplasmic shuttling of Regnase-1 is inhibited by 14-3-3's association with Reg-nase-1. Collectively, we identified a novel 14-3-3-mediated molecular mechanism which controls Regnase-1; a distinctly independent mechanism from βTRCP-mediated protein degradation of Reg-nase-1.

## Results

### Regnase-1 interactome analysis revealed dynamic recruitment of 14-3-3 upon stimulation

To comprehensively uncover Regnase-1-associating proteins in steady state and under inflammatory conditions, we stimulated HeLa cells expressing FLAG-HA-tagged Regnase-1 with or without IL-1β and immunoprecipitated Regnase-1 immediately after the treatment with a crosslinking reagent, Dithiobis(succinimidyl propionate) (DSP) (*Figure 1A*). Consistent with the previous reports, mass spectrometry analysis revealed that Regnase-1 interacted with translation-related proteins such as ribosomal proteins in unstimulated cells (*Mino et al., 2015*). Whereas IL-1β stimulation reduced the association between Regnase-1 and translation-related proteins, the stimulation strongly induced the association between Regnase-1 and SCF complex proteins such as βTRCP1/2, CUL1, and SKP1 (*Iwasaki et al., 2011*). In addition to these proteins, we identified 14-3-3 family proteins as novel Regnase-1-associating proteins under IL-1β-stimulated conditions (*Figure 1B*). Consistently, immuno-precipitation analysis revealed that endogenous Regnase-1 was co-precipitated with Myc-tagged

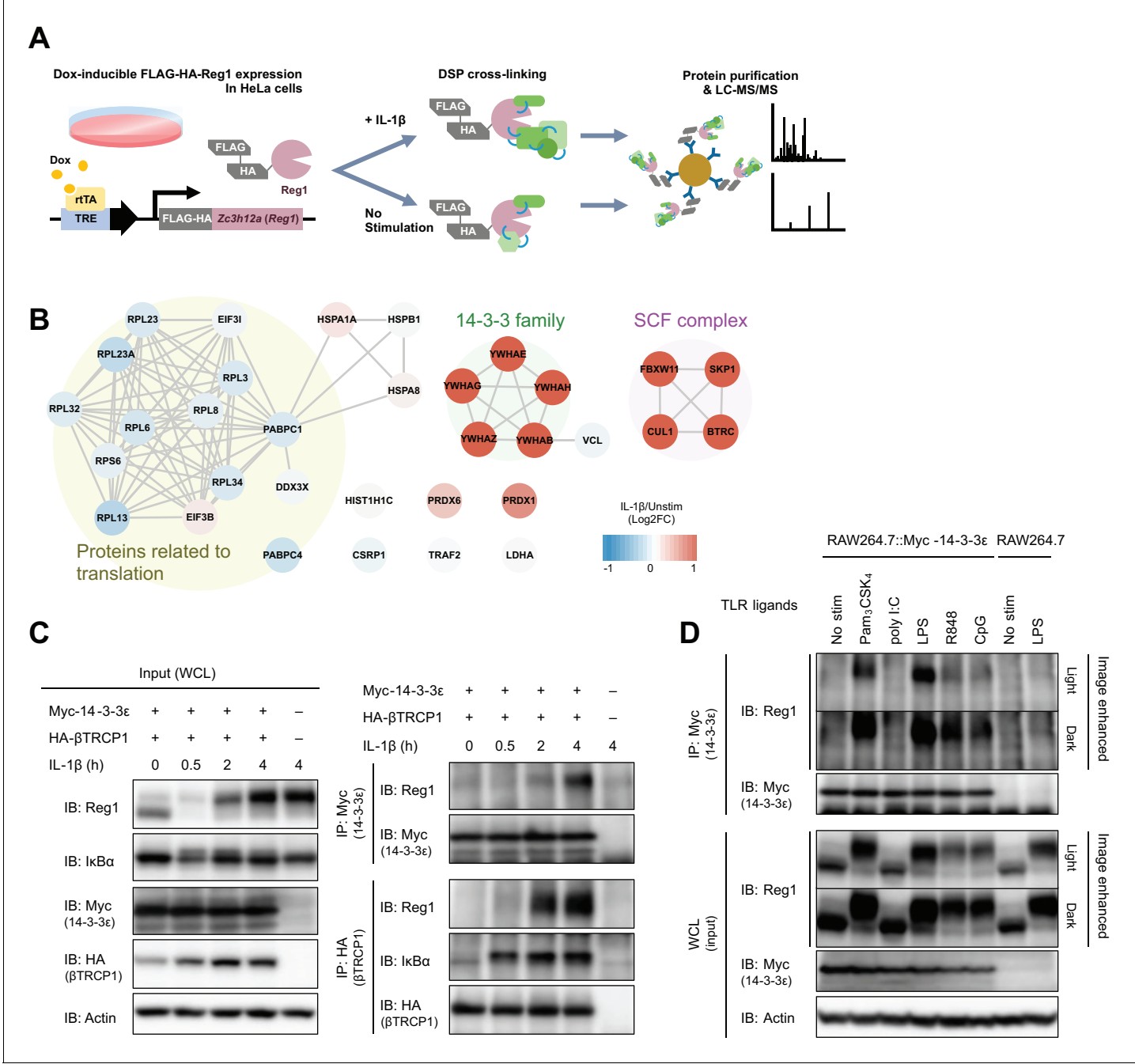

**Figure 1.** IL-1β or TLR1/2/4/7/8/9-ligand stimulation induces Regnase-1-14-3-3 interaction. (**A**) Schematic illustration of the DSP-crosslinking workflow. (**B**) Protein-protein interaction of the Regnase-1 (Reg1)-associating proteins. Each node represents Regnase-1 associating protein. The proteins whose association with Regnase-1 is weakened or enhanced in IL-1β-stimulated cells are colored in blue or red, respectively. (**C**) Immunoblot analysis of immunoprecipitates (IP: Myc or IP: HA) and WCL (whole cell lysates) from HeLa cells transiently expressing Myc-14-3-3ε and HA-βTRCP1 stimulated with IL-1β (10 ng/ml) for indicated time. (**D**) Immunoblot analysis of immunoprecipitates (IP: Myc) and WCL from RAW264.7 or RAW264.7 stably expressing Myc-14-3-3ε stimulated with Pam$_3$CSK$_4$ (10 ng/ml), poly I:C (100 μg/ml), LPS (100 ng/ml), R848 (100 nM), or CpG DNA (1 μM) for 4 hr.

The online version of this article includes the following figure supplement(s) for figure 1:

**Figure supplement 1.** Regnase-1 binds to 14-3-3 and βTRCP in response to IL-1β stimulation.

**Figure supplement 2.** Regnase-1 binds to 14-3-3β/γ/ε/ζ/η/θ but not 14-3-3σ.

14-3-3ε as well as with HA-tagged βTRCP in HeLa cells in response to IL-1β stimulation (*Figure 1C* and *Figure 1—figure supplement 1*).

As the 14-3-3 family consists of seven paralogs in human and mouse (*Aitken, 2006*), we investigated the binding of these members to Regnase-1 via immunoprecipitation (*Figure 1—figure supplement 2*). Among seven of the 14-3-3 proteins, 14-3-3-β, γ, and ε strongly interacted with Regnase-1, while 14-3-3-ζ, η, and θ showed weak interaction. Interestingly, Regnase-1 failed to associate with 14-3-3-σ, the latter of which was reported to exclusively form a homodimer but not a heterodimer with other 14-3-3 isoforms (*Verdoodt et al., 2006*).

To investigate if stimulation with TLR ligands also induces Regnase-1-14-3-3 binding, we stimulated RAW267.4 macrophages stably expressing Myc-14-3-3ε with Pam$_3$CSK$_4$ (a ligand for TLR1/2), poly I:C (a ligand for TLR3), LPS (a ligand for TLR4), R848 (a ligand for TLR7/8), or CpG DNA (a ligand for TLR9), and immunoprecipitated 14-3-3ε with an anti-Myc antibody. The Regnase-1-14-3-3 interaction was induced by all TLR ligands tested except for poly I:C (*Figure 1D*). All TLRs other than TLR3 signal through MyD88, while TLR3 utilizes TRIF as an adaptor to trigger intracellular signaling (*Fitzgerald and Kagan, 2020*; *O'Neill et al., 2013*; *Takeuchi and Akira, 2010*). Considering that IL-1β signal is also dependent on MyD88 (*Akira et al., 2006*), MyD88-dependent, but not TRIF-dependent, signaling pathways trigger the Regnase-1-14-3-3 binding.

Collectively, these results demonstrate that IL-1R/TLR stimulation induces dynamic remodeling of the Regnase-1-associating protein complex from translation machineries to SCF complexes and/or 14-3-3 proteins.

## Phosphorylation of Regnase-1 at S494 and S513 is necessary for Regnase-1-14-3-3 binding

Since 14-3-3 proteins are known to recognize phosphorylated proteins (*Muslin et al., 1996*), we investigated if 14-3-3-bound Regnase-1 is phosphorylated by inflammatory stimuli. SDS-PAGE analysis revealed that Regnase-1 band migration was slower in samples stimulated with IL-1β or TLR ligands (except for a TLR3 ligand, poly I:C) - a hallmark of Regnase-1 phosphorylation (*Figures 1C–D* and *2A*, and *Figure 2—figure supplement 1*; *Iwasaki et al., 2011*; *Tanaka et al., 2019*). Indeed, the mobility change of Regnase-1 was abolished when the cell lysates were treated with λ-protein phosphatase (λPP) (*Figure 2A–B*). Furthermore, the Regnase-1 band in the 14-3-3-precipitate migrated slower; λPP treatment of the 14-3-3-precipitate abolished this phenomenon (*Figure 2A–B*). Thus, 14-3-3 specifically binds to phosphorylated Regnase-1.

We next scrutinized Regnase-1 phosphorylation sites induced by IL-1β stimulation to identify phosphorylation sites critical for the Regnase-1-14-3-3 interaction. We purified FLAG-HA-Regnase-1 from HeLa cells stimulated with or without IL-1β and identified IL-1β-inducible phosphorylation sites by LC-MS/MS (*Figure 2C* and *Figure 2—figure supplement 2*). We found that the phosphorylation at S21, S61, S62, S362, S439, S470, S494, and S513 of Regnase-1 was increased in response to IL-1β stimulation. To identify Regnase-1 phosphorylation sites responsible for binding with 14-3-3, we mutated serine residues on Regnase-1 phosphorylation sites into alanine and probed its association with 14-3-3. Among the Regnase-1-SA mutants, S494A and S513A mutants failed to be co-precipitated with 14-3-3 (*Figure 2D*), indicating that phosphorylation at both of S494 and S513 is necessary for the Regnase-1-14-3-3 interaction. Both phosphorylation sites harbor a pSxP sequence, which shows similarity with a known 14-3-3-binding motif, RxxpSxP, mode 1 (*Yaffe et al., 1997*). Noteworthy, amino acid sequences surrounding S494 and S513 are highly conserved among many species (*Figure 2E–F*).

We next investigated the mechanism of how Regnase-1 phosphorylation is regulated by inflammatory stimuli. In response to IL-1β or TLR ligands stimulation, MyD88 associates with IRAK kinases, IRAK1 and IRAK2, via the death domain (*Gottipati et al., 2008*; *Wesche et al., 1997*). A part of C-terminal region of IRAKs in turn interacts with TRAF6 to activate NF-κB (*Ye et al., 2002*). It was shown that *Irak1* and *Irak2* double deficiency abolished the phosphorylation of Regnase-1 after LPS stimulation (*Iwasaki et al., 2011*). We found that gene depletion of *IRAK1/2* using the CRISPR-Cas9 system in HeLa cells severely impaired the association between Regnase-1 and 14-3-3 as well as the phosphorylation of Regnase-1 in response to IL-1β stimulation (*Figure 2—figure supplement 3*). Reciprocally, overexpression of IRAK1 and IRAK2-induced Regnase-1-14-3-3 binding (*Figure 2G*). In contrast, the interaction between Regnase-1 and 14-3-3 was not induced by the expression of a kinase-inactive mutant (T209A) IRAK1 (*Kollewe et al., 2004*) or a deletion mutant lacking death

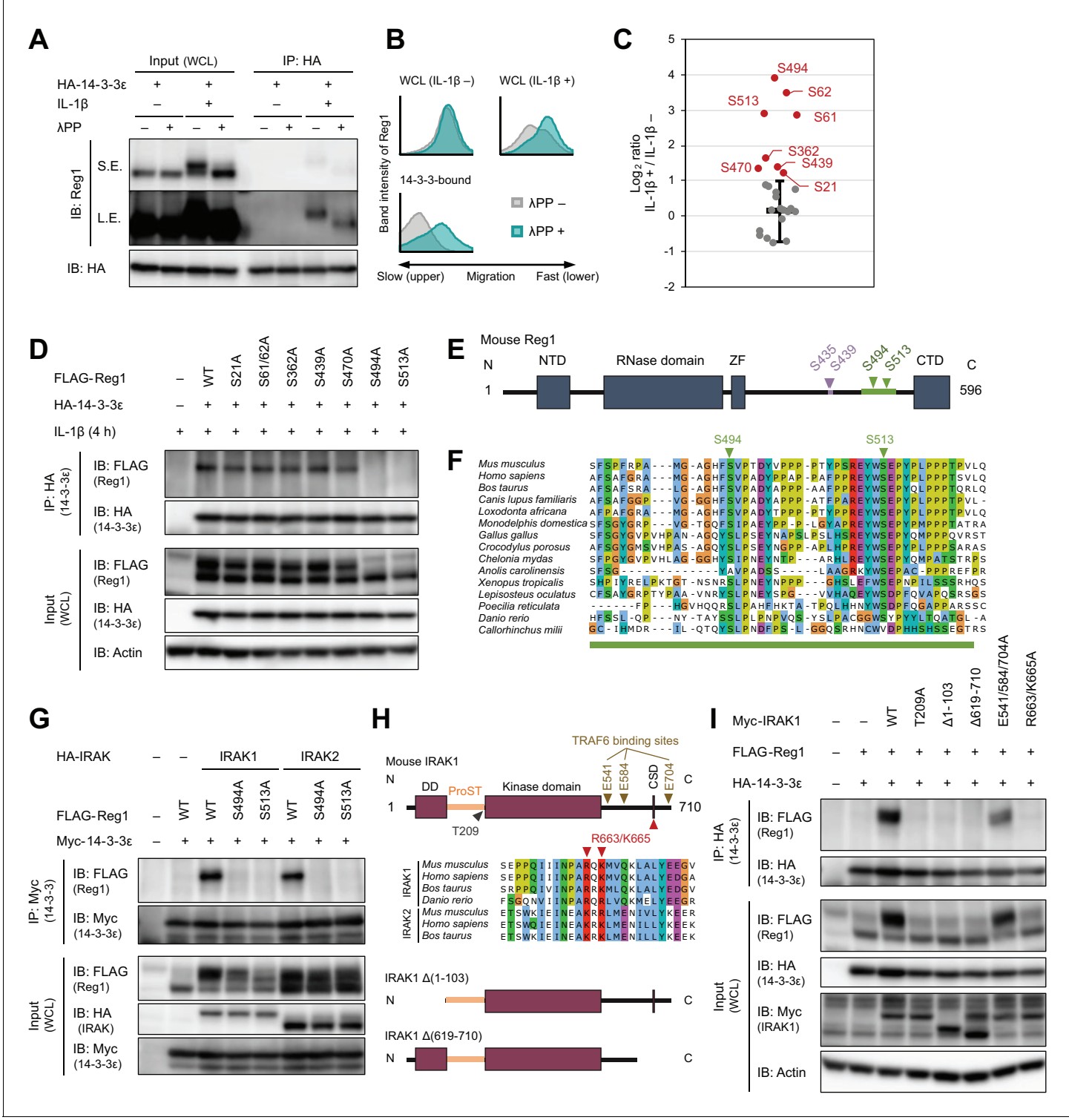

**Figure 2.** IL-1β-induced phosphorylation of Regnase-1 at S494 and S513 is necessary for Regnase-1-14-3-3 binding. (**A**) Immunoblot analysis of λPP-treated immunoprecipitates (IP: HA) and WCL from HeLa cells transiently expressing HA-14-3-3ε stimulated with IL-1β (10 ng/ml) for 4 hr. S.E.: short exposure, L.E.: long exposure. (**B**) The intensity of Regnase-1-bands in (**A**). (**C**) Quantitation of phosphosites on Regnase-1 in HeLa cells stimulated with or without IL-1β (10 ng/ml) for 4 hr. Each dot shows phosphosite quantitative ratio between IL-1β + and IL-1β -. Phosphosites with log₂ ratio > one were colored with red. Black horizontal line shows Regnase-1 protein quantitative ratio derived from the average of non-phosphopeptide quantitative ratios, and its error bars show the standard deviation. (**D**) Immunoblot analysis of immunoprecipitates (IP: HA) and WCL from HeLa cells transiently expressing HA-14-3-3ε and FLAG-Regnase-1-WT or indicated mutants stimulated with IL-1β (10 ng/ml) for 4 hr. (**E**) Schematic illustration of Regnase-1 protein. The

*Figure 2 continued on next page*

*Figure 2 continued*

amino acid sequence including S494 and S513 shown in (**F**) is highlighted in green. NTD: N-terminal domain, ZF: Zinc finger domain, CTD: C-terminal domain. (**F**) The amino acid sequences including S494 and S513 of Regnase-1 from mouse and other indicated vertebrates. (**G**) Immunoblot analysis of immunoprecipitates (IP: Myc) and WCL from HeLa cells transiently expressing Myc-14-3-3ε, HA-IRAK1/2, and FLAG-Regnase-1-WT or indicated mutants. (**H**) Schematic illustration of IRAK1 protein. The amino acid sequence in CSD of IRAK1 and IRAK2 from mouse and other indicated vertebrates are also shown. DD: Death domain, CSD: C-terminal structural domain. (**I**) Immunoblot analysis of immunoprecipitates (IP: HA) and WCL from HeLa cells transiently expressing FLAG-Regnase-1-WT, HA-14-3-3ε, and Myc-IRAK1-WT or indicated mutants.

The online version of this article includes the following figure supplement(s) for figure 2:

**Figure supplement 1.** Regnase-1 bands migrate slower in LPS-stimulated samples.
**Figure supplement 2.** Candidate spectra of Regnase-1 phosphopeptides with confident site localization.
**Figure supplement 3.** Regnase-1-14-3-3 interaction is impaired in IRAK1/2-depleted cells.
**Figure supplement 4.** Schematic illustration of IRAK1.
**Figure supplement 5.** R663/K665A mutation does not abrogate IRAK1-mediated NF-κB activation.
**Figure supplement 6.** IL-17A stimulation induces phosphorylation at S494 and S513 of Regnase-1.
**Figure supplement 7.** Candidate spectra of Regnase-1 phosphopeptides with confident site localization.
**Figure supplement 8.** IL-17A stimulation induces Regnase-1-14-3-3 association.

domain (Δ1–103) of IRAK1, indicating that the Regnase-1-14-3-3 binding requires the IRAK1 kinase activity as well as recruitment to MyD88 (*Figure 2H–I*). Although the C-terminal 619–710 portion of IRAK1 was also required for Regnase-1-14-3-3 binding, point mutations in TRAF6 binding sites (E541/E584/E704A) (*Ye et al., 2002*) did not abolish the Regnase-1-14-3-3 binding (*Figure 2H–I*). In silico prediction suggested the presence of a C-terminal structural domain (CSD) in the 619–710 of IRAK1 (*Figure 2—figure supplement 4*). In the CSD of IRAK1, highly conserved amino acids, R663 and K665, are critical for the Regnase-1-14-3-3 binding (*Figure 2H–I*), suggesting that the CSD of IRAK1 controls Regnase-1-14-3-3 interaction irrespective of the recruitment of TRAF6. Of note, the R663/K665A mutant IRAK1 was capable of activating NF-κB (*Figure 2—figure supplement 5*), indicating that the IRAK1 C-terminal region has two distinct functions: NF-κB activation through TRAF6-binding sites and the induction of Regnase-1-14-3-3 interaction through the CSD.

S494 and S513 of Regnase-1 are also reported to be phosphorylated by overexpression of Act1 together with TANK-binding kinase 1 (TBK1) or IKK-i/ε, which mimics IL-17 signaling (*Tanaka et al., 2019*). We detected phosphorylation at S494 and S513 of Regnase-1 in IL-17A-stimulated cells as well as IL-1β-stimulated cells by LC-MS/MS (*Figure 2—figure supplements 6* and *7*). Furthermore, we found that IL-17A stimulation also induced Regnase-1-14-3-3 binding (*Figure 2—figure supplement 8*).

Collectively, these data demonstrate that the IRAK-dependent phosphorylation of Regnase-1 at S494 and S513 is necessary for the association between Regnase-1 and 14-3-3.

## βTRCP binds to 14-3-3-free Regnase-1

MyD88-dependent signaling also induces IKK-mediated phosphorylation of Regnase-1 at S435 and S439, which allows recognition of Regnase-1 by βTRCP (*Iwasaki et al., 2011*). With this, we examined the relationship between the association of Regnase-1 to 14-3-3 and to βTRCP. We found that Regnase-1 harboring S435A and S439A mutations permitted the interaction with 14-3-3 but failed to recruit βTRCP (*Figure 3A–B*). Reciprocally, the S494A or S513A mutation of Regnase-1 did not inhibit the association between Regnase-1 and βTRCP (*Figure 3B*), indicating that the phosphorylation of Regnase-1 at S494 or S513 or the Regnase-1-14-3-3 binding is dispensable for the Regnase-1-βTRCP association. We next compared the phosphorylation status of βTRCP-bound and 14-3-3-bound Regnase-1. Since βTRCP-mediated polyubiquitination potentially alters the molecular weight of Regnase-1, we utilized a βTRCP mutant which is unable to induce polyubiquitination due to the lack of the F-box domain (βTRCP-ΔF). Interestingly, the SDS-PAGE analysis revealed that βTRCP-ΔF-bound Regnase-1 migrated faster than 14-3-3-bound Regnase-1 (*Figure 3C–D*), indicating that βTRCP likely binds to 14-3-3-free Regnase-1.

These results demonstrate that the binding of Regnase-1 to 14-3-3 and βTRCP occurs independently although IL-1β stimulation simultaneously induces phosphorylation of Regnase-1 at S494 and S513 as well as S435 and S439. In addition, 14-3-3 inhibits the Regnase-1-βTRCP binding.

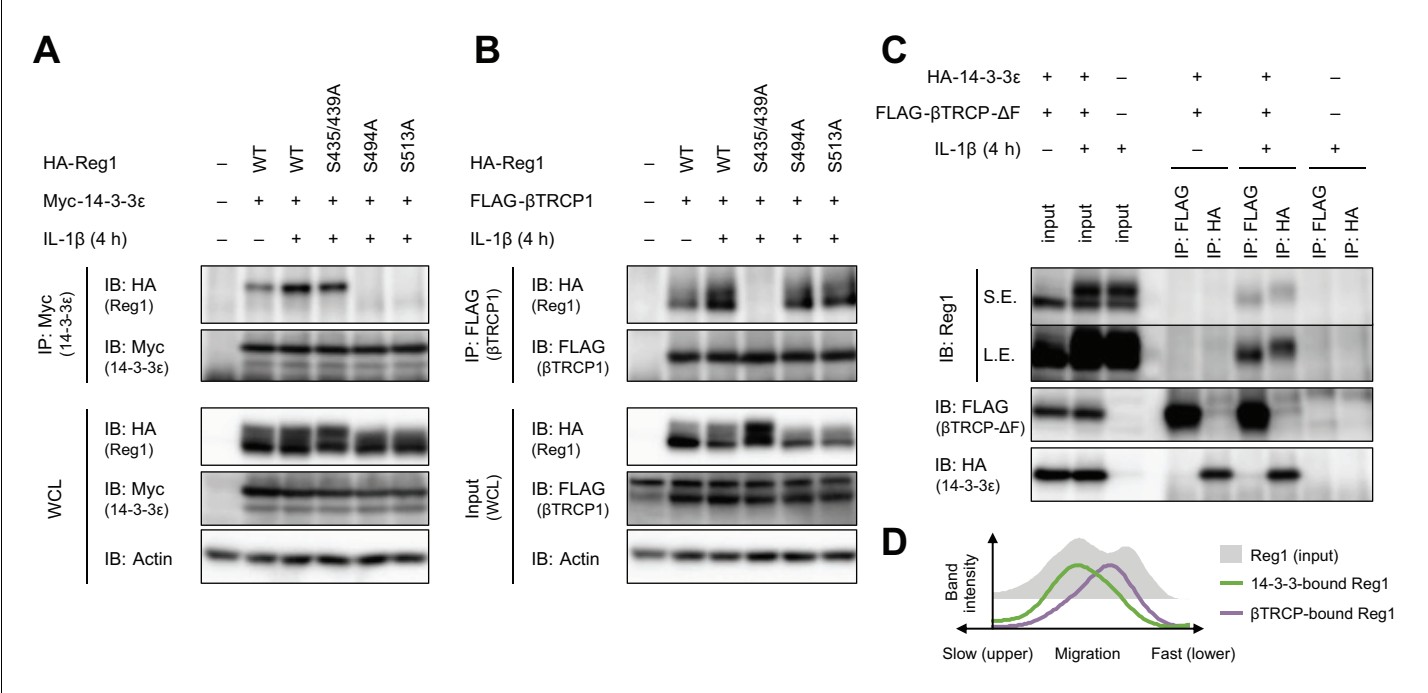

**Figure 3.** βTRCP binds to 14-3-3-free Regnase-1. (**A**) Immunoblot analysis of immunoprecipitates (IP: Myc) and WCL from HeLa cells transiently expressing Myc-14-3-3ε and HA-Regnase-1-WT or indicated mutants stimulated with IL-1β (10 ng/ml) for 4 hr. (**B**) Immunoblot analysis of immunoprecipitates (IP: FLAG) and WCL from HeLa cells transiently expressing FLAG-βTRCP1 and HA-Regnase-1-WT or indicated mutants stimulated with IL-1β (10 ng/ml) for 4 hr. (**C**) Immunoblot analysis of immunoprecipitates (IP: FLAG or HA) and WCL from HeLa cells transiently expressing FLAG-β TRCP-ΔF and HA-14-3-3ε stimulated with IL-1β (10 ng/ml) for 4 hr. S.E.: short exposure, L.E.: long exposure. (**D**) The intensity of Regnase-1-bands in (**C**).

## The S513A mutation destabilizes Regnase-1 protein without affecting target mRNA abundance

To evaluate the functional roles of Regnase-1-14-3-3 interaction, we generated $Zc3h12a^{S513A/S513A}$ knock-in mice (*Figure 4—figure supplement 1*). $Zc3h12a^{S513A/S513A}$ mice did not show gross abnormality, nor did they exhibit alteration in the numbers of T, B cells or macrophages (data not shown). We stimulated mouse embryonic fibroblasts (MEFs) derived from $Zc3h12a^{WT/WT}$ and $Zc3h12a^{S513A/S513A}$ mice with IL-1β and checked Regnase-1 expression (*Figure 4A*). Immunoblot analysis revealed that Regnase-1 was degraded 30 min after stimulation in both WT and S513A mutant MEFs. Following this, Regnase-1 levels increased in WT MEFs at 2 and 4 hr after stimulation (*Figure 4A*). Notably, most of the newly synthesized Regnase-1 showed slow migration, consistent with the immunoprecipitation experiment using HeLa cells or RAW264.7 cells shown in *Figure 1C and D*. On the other hand, the slowly migrating Regnase-1 band did not appear in $Zc3h12a^{S513A/S513A}$ MEFs after IL-1β stimulation. Interestingly, the amount of Regnase-1 at lower bands, which are not the binding target of 14-3-3 (*Figure 2A*), was comparable between WT and $Zc3h12a^{S513A/S513A}$ at corresponding time points. Consequently, total Regnase-1 protein expression was severely reduced in $Zc3h12a^{S513A/S513A}$ MEFs compared with WT after IL-1β stimulation (*Figure 4A*). Similar results were also obtained when bone marrow-derived macrophages (BMDMs) and thioglycollate-elicited peritoneal exudate cells (PECs) derived from $Zc3h12a^{WT/WT}$ and $Zc3h12a^{S513A/S513A}$ mice were stimulated with LPS (*Figure 4B–C*). Nevertheless, $Zc3h12a$ mRNA levels were comparable between $Zc3h12a^{WT/WT}$ and $Zc3h12a^{S513A/S513A}$ cells (*Figure 4D–F*), suggesting that S513A mutation affects the protein stability of Regnase-1. To address this, we examined the kinetics of Regnase-1 degradation following LPS stimulation by treating cells with cycloheximide (CHX). Indeed, Regnase-1-S513A was more rapidly degraded than Regnase-1-WT in PECs after LPS stimulation (*Figure 4—figure supplement 2*). Furthermore, treatment of $Zc3h12a^{S513A/S513A}$ PECs with MG-132, a proteasome inhibitor, resulted in the increase of smearing in the band patterns of Regnase-1 in LPS-stimulated cells (*Figure 4C*), possibly due to the inhibition of degradation of polyubiquitinated Regnase-1. These data indicate that

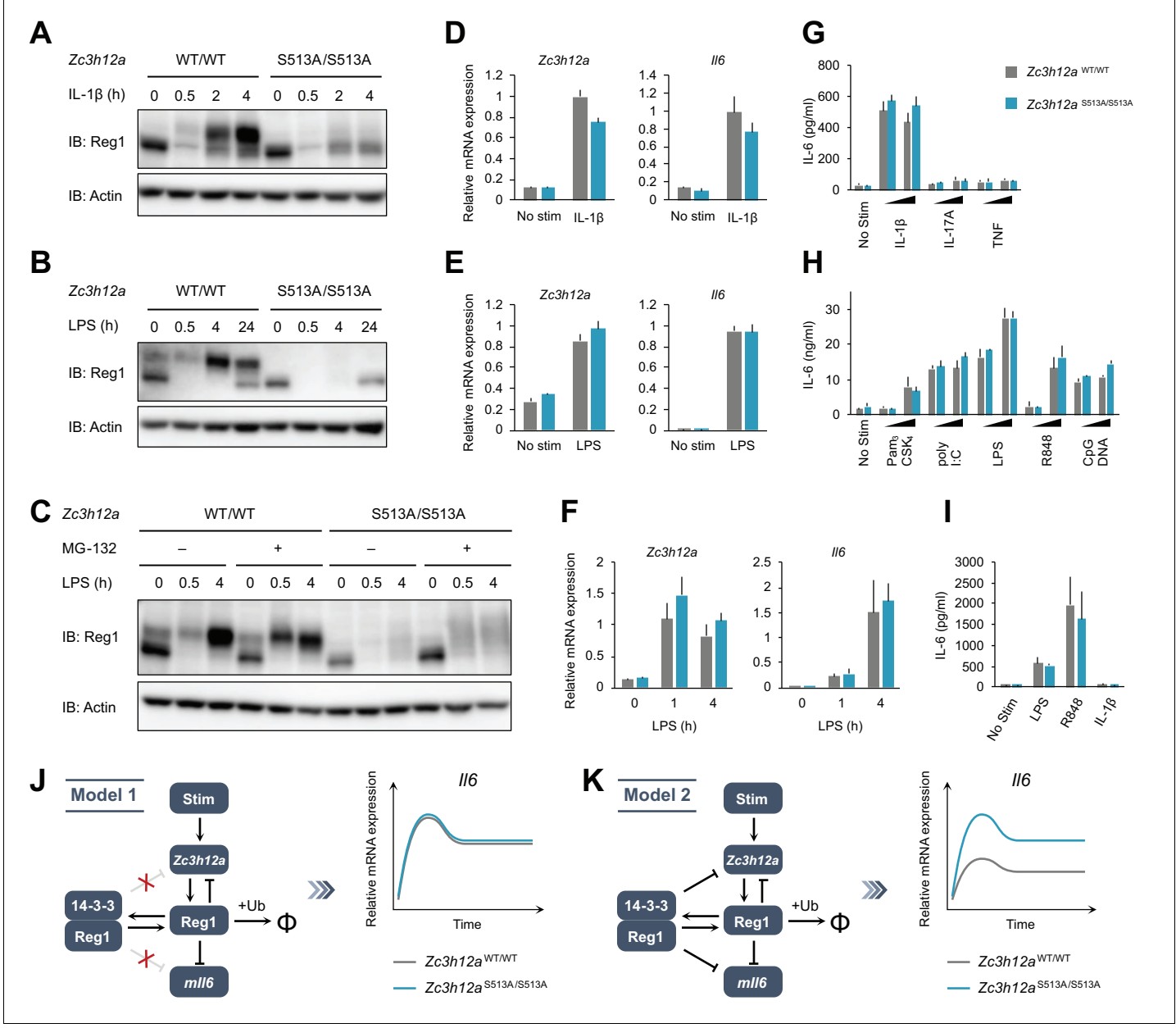

**Figure 4.** The S513A mutation destabilizes Regnase-1 protein but does not affect target mRNA abundance. (A–C) Immunoblot analysis of *Zc3h12a*[WT/WT] and *Zc3h12a*[S513A/S513A] MEFs stimulated with IL-1β (10 ng/ml) (A), BMDMs stimulated with LPS (100 ng/ml) (B), and thioglycollate-elicited PECs stimulated with LPS (100 ng/ml) (C) for indicated time. PECs were pretreated with MG-132 (5 μM) 2 hr before the stimulation. (D)-(F) mRNA expression of *Zc3h12a* and *Il6* in *Zc3h12a*[WT/WT] and *Zc3h12a*[S513A/S513A] MEFs stimulated with IL-1β (10 ng/ml) for 4 hr (D), BMDMs stimulated with LPS (100 ng/ml) for 4 hr (E), and thioglycollate-elicited PECs stimulated with LPS (100 ng/ml) for indicated time (F). (G)-(I) IL-6 secretion in *Zc3h12a*[WT/WT] and *Zc3h12a*[S513A/S513A] MEFs stimulated with IL-1β (10 ng/ml), IL-17A (50 ng/ml), or TNF (10 ng/ml) for 24 hr (G), BMDMs stimulated with Pam₃CSK₄ (1 or 10 ng/ml), poly I:C (10 or 100 μg/ml), LPS (10 or 100 ng/ml), R848 (10 or 100 nM), or CpG DNA (0.1 or 1 μM) for 24 hr (H), and thioglycollate-elicited PECs stimulated with LPS (100 ng/ml), R848 (100 nM), or IL-1β (10 ng/ml) for 24 hr (I). (J) Schematic representation of Model 1 in which 14-3-3-bound Regnase-1 does not have the function of degrading its target mRNAs. This model could explain the experimental observations. (K) Schematic representation of Model 2 in which 14-3-3-bound Regnase-1 maintains some ability to degrade its target mRNAs. This model is not consistent with the experimental observations. In (D)-(I), bars represent mean values of biological replicates (*n* = 3), and error bars represent standard deviation. Data is representative of two independent experiments, each with three biological replicates.

The online version of this article includes the following figure supplement(s) for figure 4:

**Figure supplement 1.** Schematic illustration of *Zc3h12a* gene in mice.

**Figure supplement 2.** The protein stability of Regnase-1-WT and S513A.

*Figure 4 continued on next page*

*Figure 4 continued*

**Figure supplement 3.** *Il6* expression in *Zc3h12a*[−/−] PECs mRNA expression of *Il6* and *Zc3h12a* in *Zc3h12a*[WT/WT] and *Zc3h12a*[−/−] thioglycollate-elicited PECs stimulated with LPS (100 ng/ml) for indicated time.
**Figure supplement 4.** The stability of Regnase-1 target mRNAs.
**Figure supplement 5.** S513A mutation of Regnase-1 does not affect gene expression.

the phosphorylation of Regnase-1 at S513 stabilizes Regnase-1 protein after IL-1β or LPS stimulation by binding with 14-3-3.

We next checked whether the altered Regnase-1 expression by the S513A mutation affects Regnase-1-mediated mRNA decay. Despite the huge difference in Regnase-1 expression, the expression of *Il6*, a transcript degraded by Regnase-1 (*Figure 4—figure supplement 3*), was comparable between *Zc3h12a*[WT/WT] and *Zc3h12a*[S513A/S513A] cells (*Figure 4D–I*). In addition, the stability of Regnase-1 target mRNAs including *Il6*, *Zc3h12a*, and *Nfkbiz* was equivalent between *Zc3h12a*[WT/WT] and *Zc3h12a*[S513A/S513A] cells (*Figure 4—figure supplement 4*). Furthermore, even when we analyzed gene expression profile comparing *Zc3h12a*[WT/WT] and *Zc3h12a*[S513A/S513A] macrophages by an RNA-seq analysis (*Figure 4—figure supplement 5*), we did not identify any differentially expressed genes (adj p<0.05) between *Zc3h12a*[WT/WT] and *Zc3h12a*[S513A/S513A] macrophages irrespective of the stimulation with LPS.

To examine the mechanisms underlying these observations, we developed two mathematical models based on our previous studies (see Materials and methods) (*Iwasaki et al., 2011*; *Mino et al., 2019*). The first model (Model 1) assumes that 14-3-3-bound Regnase-1 is unable to degrade its target mRNAs (*Figure 4J*). The second model (Model 2) assumes that Regnase-1 binding with 14-3-3 maintains its ability to degrade its target mRNAs to a certain extent (*Figure 4K*). Mathematical analysis showed that in Model 2, the abundance of the *Il6* mRNAs should be different between *Zc3h12a*[WT/WT] and *Zc3h12a*[S513A/S513A] cells under the condition that the amount of 14-3-3-free Regnase-1 protein (lower bands in *Figure 4A–C*) is comparable between them. Our observations that the abundance of the target mRNAs did not differ between *Zc3h12a*[WT/WT] and *Zc3h12a*[S513A/S513A] cells in the late phase of stimulation is inconsistent with Model 2, suggesting that Regnase-1 is inactivated upon binding to 14-3-3.

These results imply that the phosphorylation at S513 and the following association with 14-3-3 nullifies Regnase-1's ability in degrading target mRNAs, although it stabilizes and significantly upregulates the abundance of Regnase-1.

## 14-3-3 inactivates Regnase-1 by inhibiting Regnase-1-RNA binding

The mathematical analysis suggests that 14-3-3-bound Regnase-1 is inactive as the S513A mutation failed to affect *Il6* expression in MEFs or macrophages. To examine if this comparable *Il6* expression was due to increased degradation of Regnase-1-S513A protein via βTRCP, we further mutated βTRCP-recognition sites, S435 and S439, to alanine in Regnase-1-S513A (Regnase-1-S435/439/513A). As shown in *Figure 5A*, the Regnase-1-S435/439/513A mutant was more abundantly expressed than Regnase-1-S513A after IL-1β stimulation, indicating that Regnase-1-S513A is degraded via the association with βTRCP. It is noteworthy that most of Regnase-1-S435/439/513A showed fast migration, whereas the majority of Regnase-1-S435/439A migrated slowly in response to the stimulation. To verify if 14-3-3-bound Regnase-1 is functional or not, we assessed the target mRNA suppression activity of each mutant by checking the expression of *Il6* co-transfected with Regnase-1. Regnase-1-S435/439/513A was more potent in suppressing *Il6* expression compared to WT or other SA mutants, S513A and S435/439A, in response to IL-1β stimulation (*Figure 5B*). These results indicate that IL-1β stimulation regulates the activity of Regnase-1 by two independent mechanisms via 14-3-3 and βTRCP, respectively.

To further examine the mechanism of how 14-3-3 inactivates Regnase-1, we attempted to generate a Regnase-1 mutant which constitutively binds to 14-3-3 even without stimulation. We generated a phospho-mimic mutant of Regnase-1 (S494D/S513D). However, this mutant failed to bind 14-3-3 (*Figure 5C*), indicating that the phosphate moiety, but not negative charge, is recognized by 14-3-3. Then we utilized a sequence of Exoenzyme S (ExoS), which is a bacterial protein derived from *Pseudomonas aeruginosa* and is known to bind to 14-3-3 without phosphorylation (*Fu et al., 1993*; *Karlberg et al., 2018*; *Masters et al., 1999*). The 22 amino acids of Regnase-1 covering S494 and

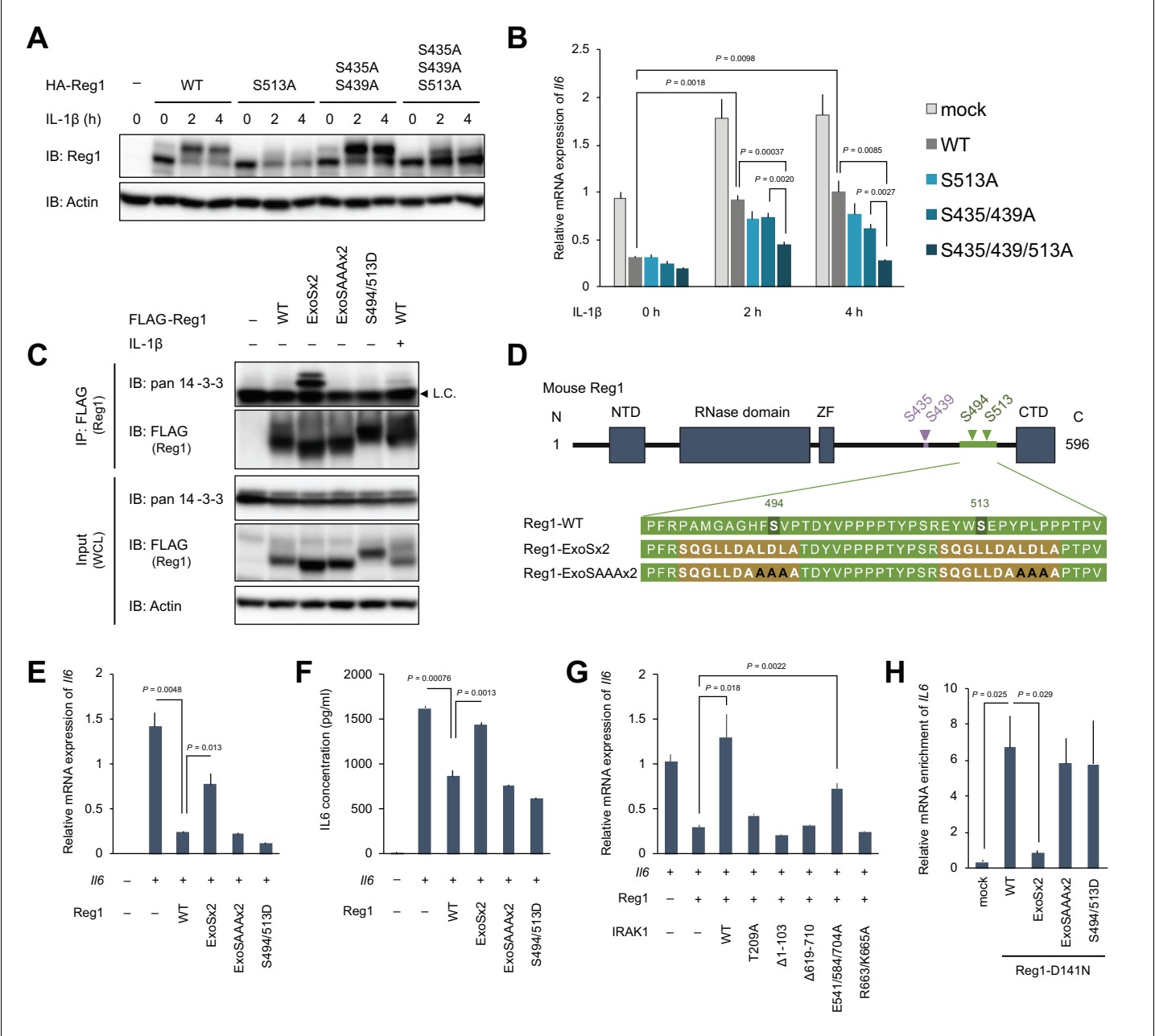

**Figure 5.** 14-3-3 bound to phospho-S494 and S513 inactivates Regnase-1 by inhibiting Regnase-1-mRNA binding. (A) Immunoblot analysis of *ZC3H12A*-KO HeLa cells transiently expressing Regnase-1-WT or indicated mutants. Cells were stimulated with IL-1β (10 ng/ml) for indicated time. (B) mRNA expression of *Il6* in HeLa cells transiently expressing Regnase-1-WT or indicated mutants together with *Il6*. Cells were stimulated with IL-1β (10 ng/ml) for indicated time. (C) Immunoblot analysis of immunoprecipitates (IP: FLAG) and WCL from HeLa cells transiently expressing FLAG-Regnase-1-WT or indicated mutants. For the IL-1β stimulation, cells were stimulated with IL-1β (10 ng/ml) for 4 hr. L.C.: light chain. (D) Schematic illustration of Regnase-1 and the amino acid sequences of Regnase-1-WT, -ExoSx2, and ExoSAAAx2. NTD: N-terminal domain, ZF: Zinc finger domain, CTD: C-terminal domain. (E) mRNA expression of *Il6* in HeLa cells transiently expressing Regnase-1-WT or indicated mutants together with *Il6*. (F) Secreted IL6 concentration in (E). (G) mRNA expression of *Il6* in HeLa cells transiently expressing Regnase-1-WT and IRAK1-WT or indicated mutants together with *Il6*. (H) The amount of *IL6* mRNAs immunoprecipitated with FLAG-Regnase-1-D141N or other indicated mutants in HeLa cells. In (B), (E)-(H), bars represent mean values of biological replicates (*n* = 3), and error bars represent standard deviation. p-Values were calculated using unpaired, two-sided t-test. Data is representative of two independent experiments, each with three biological replicates.

The online version of this article includes the following figure supplement(s) for figure 5:

**Figure supplement 1.** Regnase-1-ExoSx2-D141N binds to 14-3-3.

**Figure supplement 2.** Regnase-1-ExoSx2-D141N failed to bind to target mRNAs.

S513 were substituted with two ExoS (419-429) sequences (*Figure 5D*). As a control, we additionally mutated Regnase-1-ExoSx2 by substituting its core sequences for 14-3-3 binding (L426, D427, and L428) with alanine residues (Regnase-1-ExoSAAAx2) (*Ottmann et al., 2007*; *Yasmin et al., 2006*). We observed that Regnase-1-ExoSx2, but not Regnase-1-ExoSAAAx2, interacted with endogenous 14-3-3 without any stimulation (*Figure 5C*). Using these mutants, we investigated whether 14-3-3 binding alters the activity of Regnase-1 to suppress *Il6* expression. Consistent with its 14-3-3 binding capacity, Regnase-1-ExoSx2, but not Regnase-1-ExoSAAAx2 and -S494D/S513D, lost the activity to inhibit *Il6* expression (*Figure 5E*). Furthermore, the production of IL-6 protein was similarly inhibited depending on the capacity of Regnase-1 to bind 14-3-3 (*Figure 5F*). In addition, Regnase-1-mediated suppression of *Il6* expression was impaired by the overexpression of IRAK1-WT and E541/E584/E704A mutants, both of which induce Regnase-1-14-3-3 association (*Figure 5G*). On the other hand, IRAK1 mutants that failed to induce the Regnase-1-14-3-3 association (T209A, Δ1–103, Δ619–710, and R663/K665A) did not affect the activity of Regnase-1.

We next examined how 14-3-3 inhibits the activity of Regnase-1 by investigating Regnase-1-mRNA binding activity using various Regnase-1 mutants in HeLa cells. To stabilize Regnase-1-RNA binding, we generated a nuclease inactive version of Regnase-1 by introducing the D141N mutation to each of Regnase-1 mutant (*Matsushita et al., 2009*; *Figure 5—figure supplement 1*). As shown in *Figure 5H*, forced interaction of Regnase-1-D141N with 14-3-3 by the ExoSx2 mutation in HeLa cells abrogated the binding with *IL6* mRNA, whereas *IL6* was co-precipitated with Regnase-1-D141N, -ExoSAAAx2-D141N and -S494D/S513D-D141N (*Figure 5H*). In addition to *IL6*, Regnase-1-ExoSx2 failed to bind to other reported target mRNAs such as *NFKBIZ*, *PTGS2*, and CXC chemokines (*Figure 5—figure supplement 2*). Collectively, these data demonstrate that 14-3-3 inhibits Regnase-1-mRNA binding, thereby abrogating Regnase-1-mediated mRNA degradation.

## 14-3-3 inhibits nuclear import of Regnase-1

We have previously shown that Regnase-1 interacts with CBP80-bound, but not eIF4E-bound, mRNAs (*Mino et al., 2019*), indicating that Regnase-1 degrades mRNAs immediately after the export from the nucleus to the cytoplasm (*Maquat et al., 2010*; *Müller-McNicoll and Neugebauer, 2013*). Although Regnase-1 mainly localizes in the cytoplasm (*Mino et al., 2015*), we hypothesized Regnase-1 shuttles between the nucleus and the cytoplasm to recognize its target mRNAs in association with their nuclear export. To test this hypothesis, we examined the subcellular localization of Regnase-1 following the treatment with Leptomycin B (LMB), which inhibits CRM1 (also known as Exportin-1)-mediated protein export from the nucleus to the cytoplasm (*Yashiroda and Yoshida, 2003*). Although Regnase-1 localized in the cytoplasm in the steady state condition, LMB treatment induced rapid accumulation of Regnase-1 in the nucleus within 30 min (*Figure 6A*). These results suggest that Regnase-1 dynamically changes its localization between the cytoplasm and the nucleus. Given that Regnase-1 dominantly localizes in the cytoplasm in the steady state conditions, the frequency of its nuclear export seems to be higher than its nuclear import.

CRM1 is known to recognize a nuclear export signal (NES) of a cargo protein for the protein export (*Hutten and Kehlenbach, 2007*). Thus, we investigated if Regnase-1 harbors a NES. In silico prediction deduced amino acids 433–447 of Regnase-1 as a potential NES with high probability (*Xu et al., 2015*; *Figure 6B–D*). Indeed, Regnase-1 lacking 422–451 spontaneously accumulated in the nucleus (*Figure 6E*). Since NESs are characterized by hydrophobic residues (*la Cour et al., 2003*), we also inspected which hydrophobic resides of Regnase-1 were important for the efficient nuclear export of Regnase-1. We found that L440, M444, L447, and W448 of Regnase-1 were critical for the nuclear export of Regnase-1 (*Figure 5E*). Noteworthy, all the residues are highly conserved among species (*Figure 5D*).

We next examined whether 14-3-3 binding controls the localization of Regnase-1. Interestingly, Regnase-1-ExoSx2 failed to accumulate in the nucleus even after LMB treatment while Regnase-1-WT and -ExoSAAAx2 accumulated in the nucleus by LMB treatment (*Figure 6F*). This result indicates that Regnase-1-ExoSx2 is unable to translocate into the nucleus like Regnase-1-WT. Taken together, 14-3-3 inhibits the nuclear import of Regnase-1 as well as its binding to target mRNAs.

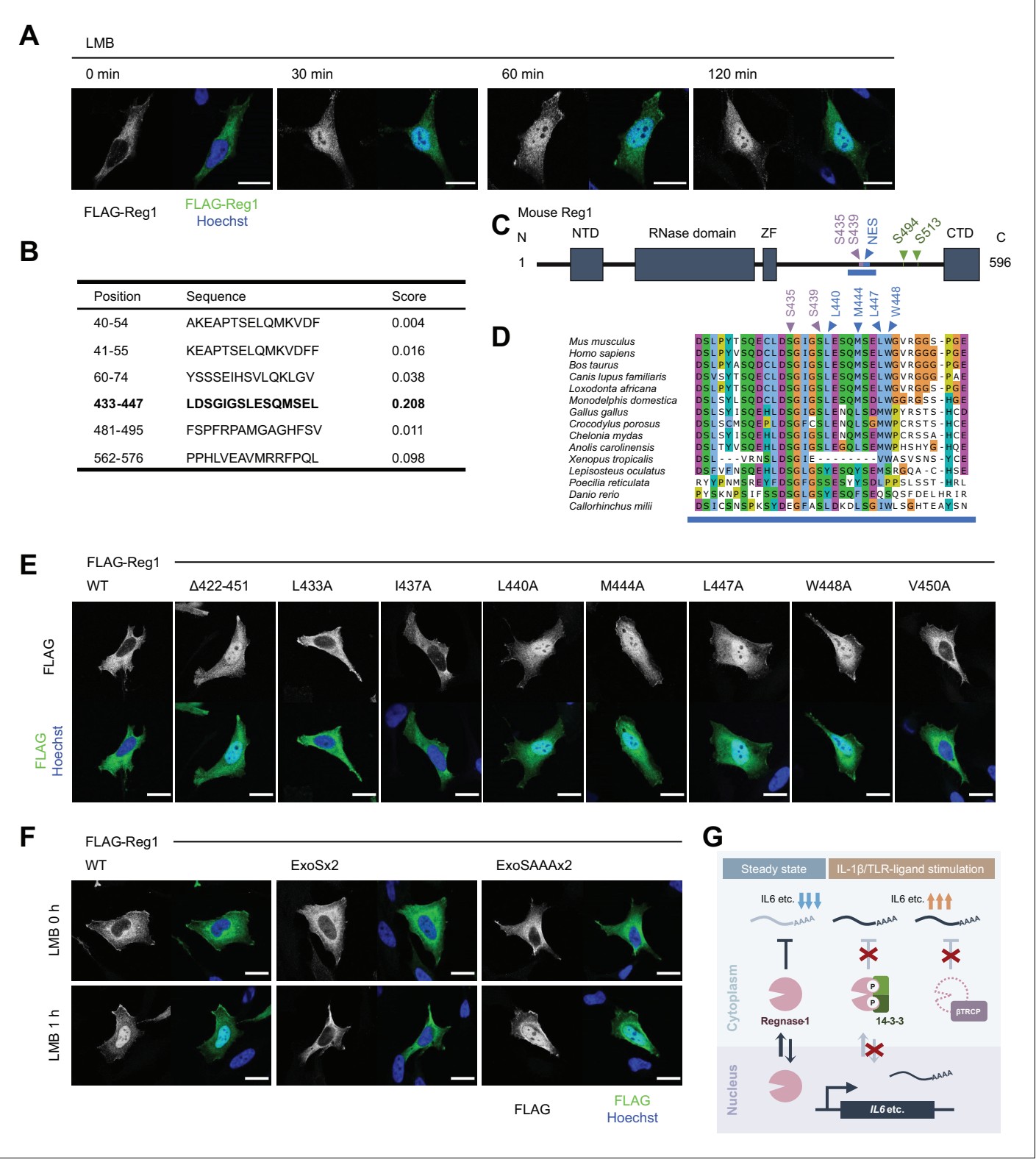

**Figure 6.** 14-3-3 inhibit nuclear-cytoplasmic shuttling of Regnase-1. (**A**) Immunofluorescence analysis of HeLa cells transiently expressing FLAG-Regnase-1-WT treated with Leptomycin B (LMB) (10 ng/ml) for indicated time. (**B**) The result of NES prediction of Regnase-1 by LocNES. Higher score indicates higher probability. (**C**) Schematic illustration of Regnase-1. The amino acid sequence shown in (**D**) is highlighted in blue. NTD: N-terminal domain, ZF: Zinc finger domain, CTD: C-terminal domain. (**D**) The amino acid sequences including S435/S439 and NES of Regnase-1 from mouse and other indicated vertebrates. (**E**) Immunofluorescence analysis of HeLa cells transiently expressing FLAG-Regnase-1-WT or indicated mutants. (**F**)

*Figure 6 continued on next page*

*Figure 6 continued*

Immunofluorescence analysis of HeLa cells transiently expressing FLAG-Regnase-1-WT or indicated mutants treated with LMB (10 ng/ml) for 1 hr. (**G**) Model of 14-3-3- and βTRCP-mediated regulation of Regnase-1. In the steady state, Regnase-1 shuttles between the nucleus and the cytoplasm and degrades target mRNAs such as *Il6*. Under IL-1β or TLR-ligands stimulation, two different regulatory mechanisms suppress the activity of Regnase-1 not to disturb proper expression of inflammatory genes; βTRCP induces protein degradation of Regnase-1 and 14-3-3 inhibits nuclear-cytoplasmic shuttling and mRNA recognition of Regnase-1. In (**A**), (**E**), and (**F**), white scale bars indicate 20 μm.

## Discussion

In the present study, we discovered that IL-1β and TLR stimulation dynamically changes protein-protein interaction of Regnase-1. Particularly, these stimuli trigger the interaction of Regnase-1 with 14-3-3 as well as βTRCP via phosphorylation at distinct amino acids. Whereas phosphorylation of Regnase-1 at S494 and S513 is recognized by 14-3-3, βTRCP associates with Regnase-1 phosphorylated at S435 and S439. We demonstrated that each Regnase-1-14-3-3 and Regnase-1-βTRCP binding occurs independently. Furthermore, 14-3-3 prevent Regnase-1-βTRCP binding, resulting in protein stabilization of 14-3-3-bound Regnase-1.

The amino acid sequence surrounding S494 and S513 of Regnase-1 is widely conserved among species. Particularly, both S494 and S513 harbor the pSxP sequence motif, which overlaps with a known 14-3-3 binding motif, RxxpSxP, mode 1 (*Yaffe et al., 1997*). It is quite plausible that the 14-3-3 dimer directly binds to phosphorylated S494 and S513 of Regnase-1 with the phospho-peptide binding groove. Interestingly, only 14-3-3σ, which forms a homodimer but not a heterodimer with other 14-3-3 isoforms (*Verdoodt et al., 2006*), failed to bind with Regnase-1 (*Figure 1—figure supplement 2*). This result might be a clue to elucidate the target specificity of each 14-3-3 paralog.

14-3-3 and βTRCP inhibit Regnase-1-mediated mRNA decay via distinct mechanisms; 14-3-3 prevents Regnase-1-mRNA binding while βTRCP induces protein degradation of Regnase-1. Analysis of *Zc3h12a*[S513A/S513A] mice revealed that 14-3-3-mediated abrogation of Regnase-1 can be compensated by the degradation of Regnase-1. The presence of this dual regulatory system underscores the importance of restricting the activity of Regnase-1 to ensure proper inflammatory gene expression when cells encounter PAMPs or DAMPs (*Figure 6G*).

Notably, exome sequence analysis of the colon samples from ulcerative colitis patients discovered mutations in the βTRCP binding site of Regnase-1 (*Kakiuchi et al., 2020*; *Nanki et al., 2020*). Furthermore, a previous report showed that *Zc3h12a*[S435/S439A] mutant mice were resistant to experimental autoimmune encephalomyelitis (EAE) (*Tanaka et al., 2019*). All these mutations abolish βTRCP-mediated degradation of Regnase-1. However, genetic association between the 14-3-3-binding site of Regnase-1 and inflammatory diseases has not been identified so far. This is possibly because of the compensation by βTRCP-mediated regulation, which we observed in *Zc3h12a*[S513A/S513A] mice and HeLa cells transiently expressing Regnase-1-S513A and S435/439/513A. Previous studies have shown that viral proteins or lncRNAs inhibit βTRCP-mediated protein degradation (*Guo et al., 2020*; *Neidel et al., 2019*; *van Buuren et al., 2014*; *Yang et al., 2020*). 14-3-3-mediated regulation of Regnase-1 may serve as a backup mechanism to control the activity of Regnase-1 when βTRCP-mediated protein degradation is dysregulated.

While βTRCP regulates the abundance of Regnase-1 through protein degradation, 14-3-3 regulates the activity of Regnase-1. We found that 14-3-3-bound Regnase-1 failed to associate with mRNAs, indicating that 14-3-3 prevents Regnase-1 from recognizing target mRNA. We have previously shown that an RNase domain and an adjacent zinc finger domain play an important role in Regnase-1-RNA binding (*Yokogawa et al., 2016*). However, the 14-3-3-binding site of Regnase-1 is in the C-terminal part of Regnase-1, which is distant from RNase and zinc finger domains. Therefore, 14-3-3 is unlikely to inhibit Regnase-1-mRNA binding by simple competition between 14-3-3 and mRNAs for the RNA binding domain of Regnase-1. We have previously reported that Regnase-1 interacts with CBP80-bound, but not eIF4E-bound, mRNAs, indicating that Regnase-1 recognizes its target mRNA before or immediately after the nuclear export of the mRNA (*Mino et al., 2019*). In this study, we found that Regnase-1 shuttles between the nucleus and the cytoplasm while 14-3-3-bound Regnase-1 cannot enter the nucleus. Thus, it is tempting to speculate that Regnase-1 recognizes mRNA in the nucleus and induce mRNA decay during pioneer rounds of translation immediately after the nuclear export (*Maquat et al., 2010*; *Müller-McNicoll and Neugebauer, 2013*).

Nevertheless, further investigation is required to clarify the mechanisms of Regnase-1-mediated mRNA decay depending on its nuclear-cytoplasmic shuttling. In addition, further studies are also necessary to clarify the molecular mechanisms how 14-3-3 controls the nuclear-cytoplasmic shuttling of Regnase-1 as well as how it regulates Regnase-1-mRNA binding by exploiting systems beyond the ExoS sequence-mediated interaction with 14-3-3.

βTRCP is likely to recognize 14-3-3-free Regnase-1, indicating that 14-3-3 inhibits Regnase-1-β TRCP interaction. There are two possible mechanisms to explain this. One posits that 14-3-3 bound to phosphorylated S494 and S513 of Regnase-1 conceals βTRCP-binding site (pS435 and pS439), although the 14-3-3-binding site does not overlap with βTRCP-binding site completely. The other possible explanation is that 14-3-3-mediated inhibition of nuclear-cytoplasmic shuttling of Regnase-1 controls βTRCP-mediated Regnase-1 degradation. Indeed, βTRCP localizes not only in the cytoplasm, but also in the nucleus (*Davis et al., 2002*). It is plausible that 14-3-3 prevents Regnase-1-β TRCP interaction in the nucleus, by inhibiting nuclear-cytoplasmic shuttling of Regnase-1. Of note, the NES of Regnase-1 is located just adjacent to βTRCP-binding site (*Figure 6C–D*), implying possibility of competitive binding of βTRCP and CRM1 to Regnase-1.

Among the molecules involved in MyD88-dependent signaling, we found that IRAK1/2 are potent inducers of the interaction between Regnase-1 and 14-3-3, thereby abrogating Regnase-1-mediated mRNA decay. We also found that kinase-inactive IRAK1 failed to induce the Regnase-1-14-3-3 complex, suggesting that the kinase activity of IRAK1 is required for the phosphorylation of Regnase-1 at S494 and S513. However, previously identified substrate sequence motifs of IRAK1, pSxV, and KxxxpS (*Sugiyama et al., 2019*) do not match the sequence surrounding S494 and S513 of Regnase-1 (*Figure 2F*). Although the motif analysis does not exclude the possibility of direct phosphorylation of Regnase-1 by IRAK1, it is possible that kinases activated by IRAK1/2 phosphorylates Regnase-1 at S494 and S513.

IRAKs are involved in stabilization of inflammatory mRNAs as well as NF-κB activation (*Flannery et al., 2011*; *Hartupee et al., 2008*; *Wan et al., 2009*). A previous study showed that IRAK1-mediated mRNA stabilization does not require IRAK1-TRAF6 association (*Hartupee et al., 2008*). Interestingly, the IRAK1-TRAF6 association is also dispensable for the Regnase-1-14-3-3 binding. Instead, other evolutionarily conserved amino acids in the C-terminal structural domain (CSD) of IRAK1, R663, and K665, are critical for Regnase-1-14-3-3 binding. Considering 14-3-3-mediated inactivation of Regnase-1, it is plausible that the CSD of IRAK1 is the key for stabilization of inflammatory mRNAs.

In summary, Regnase-1 interactome analysis revealed dynamic 14-3-3-mediated regulation of Regnase-1 in response to IL-1β and TLR stimuli. Since recent studies identified Regnase-1 as a high-potential therapeutic target in various diseases (*Kakiuchi et al., 2020*; *Nanki et al., 2020*; *Wei et al., 2019*), our findings may help maximize the effect of Regnase-1 modulation or provide an alternative way to control the activity of Regnase-1.

# Materials and methods

## Key resources table

| Reagent type (species) or resource | Designation | Source or reference | Identifiers | Additional information |
|---|---|---|---|---|
| Gene (*Mus musculus*) | *Zc3h12a* | NA | Gene ID: 230738 | |
| Gene (*Homo sapiens*) | *ZC3H12A* | NA | Gene ID: 80149 | |
| Strain, strain background (*Mus musculus*) | *Zc3h12a*^WT/WT | CLEA Japan | C57BL/6 | C57BL/6JJcl |
| Strain, strain background (*Mus musculus*) | *Zc3h12a*^-/- | https://doi.org/10.1038/nature07924 | | |
| Strain, strain background (*Mus musculus*) | *Zc3h12a*^S513A/S513A | this paper | | generated using CRISPR-Cas9 system |
| Sequence-based reagent | DNA oligo (for pX330) | this paper | CACCGCGGCTCAGACCAGTACTCTC | for *Zc3h12a*^S513A/S513A generation |

*Continued on next page*

*Continued*

| Reagent type (species) or resource | Designation | Source or reference | Identifiers | Additional information |
|---|---|---|---|---|
| Sequence-based reagent | DNA oligo (for pX330) | this paper | AAACGAGAGTACTGGTCTGAGCCGC | for $Zc3h12a^{S513A/S513A}$ generation |
| Sequence-based reagent | Donor single strand oligo | this paper | GAAGGACAGGAGTGGGTGGGGGTAATGGGTACGGCTCAGCCCAGTACTCTCTGGATGGGTAGGTGGGTGGCGGGGGCACA | for $Zc3h12a^{S513A/S513A}$ generation |
| Sequence-based reagent | DNA oligo (for pX459-*IRAK1*KO) | https://doi.org/10.1093/bioinformatics/btu743 | CACCGGTCTGGTCGCGCACGATCA | |
| Sequence-based reagent | DNA oligo (for pX459-*IRAK1*KO) | https://doi.org/10.1093/bioinformatics/btu743 | AAACTGATCGTGCGCGACCAGACC | |
| Sequence-based reagent | DNA oligo (for pX459-*IRAK2*KO) | https://doi.org/10.1038/nbt.3437 | CACCGAAAACCGCAAAATCAGCCAG | |
| Sequence-based reagent | DNA oligo (for pX459-*IRAK2*KO) | https://doi.org/10.1038/nbt.3437 | AAACCTGGCTGATTTTGCGGTTTTC | |
| Antibody | Anti-mouse-Regnase-1 (rabbit polyclonal) | MBL life science | | custom antibody production (1:1000) |
| Antibody | Anti-human-Regnase-1 (rabbit polyclonal) | Atlas Antibodies | Cat # HPA032053 | (1:500) |
| Antibody | Anti-14-3-3 (pan) (mouse monoclonal) | Santa Cruz Biotechnology | Cat # sc-1657 | (1:1000) |
| Antibody | Anti-IκB-α (rabbit polyclonal) | Santa Cruz Biotechnology | Cat # sc-371 | (1:1000) |
| Antibody | Anti-IRAK1 (mouse monoclonal) | Santa Cruz Biotechnology | Cat # sc-5288 | (1:500) |
| Antibody | Anti-FLAG (mouse monoclonal) | Sigma | Cat # F3165 | (WB:1:2000, IF:1:5000) |
| Antibody | Anti-FLAG (rabbit polyclonal) | Sigma | Cat # F7425 | (1:2000) |
| Antibody | Anti-HA (mouse monoclonal) | Sigma | Cat # H3663 | (1:2000) |
| Antibody | Anti-HA (rabbit polyclonal) | Sigma | Cat # H6908 | (1:2000) |
| Antibody | Anit-Myc (mouse monoclonal) | Sigma | Cat # M4439 | (1:2000) |
| Antibody | Anti-Myc (rabbit polyclonal) | Sigma | Cat # C3956 | (1:2000) |
| Antibody | Anti-β-Actin-HRP (mouse monoclonal) | Santa Cruz Biotechnology | Cat # sc-47778-HRP | (1:2000) |
| Antibody | Anti-Mouse IgG-HRP F(ab')2 (sheep polyclonal) | cytiva | Cat # NA9310-1ML | (1:5000) |
| Antibody | Anti-Rabbit IgG-HRP F(ab')2 (donkey polyclonal) | cytiva | Cat # NA9340-1ML | (1:5000) |
| Antibody | F(ab')2-anti-Mouse IgG (H+L)-AF488 (Goat polyclonal) | Invitrogen | Cat # A11017 | (1:2000) |
| Recombinant DNA reagent | pX330-U6-Chimeric_BB-CBh-hSpCas9 | Addgene | RRID:Addgene_42230 | |
| Recombinant DNA reagent | pSpCas9(BB)-2A-Puro (PX459) V2.0 | Addgene | RRID:Addgene_62988 | |

*Continued on next page*

*Continued*

| Reagent type (species) or resource | Designation | Source or reference | Identifiers | Additional information |
|---|---|---|---|---|
| Recombinant DNA reagent | pMD2.G | Addgene | RRID:Addgene_12259 | |
| Recombinant DNA reagent | pMDLg/pRRE | Addgene | RRID:Addgene_12251 | |
| Recombinant DNA reagent | pRSV-Rev | Addgene | RRID:Addgene_12253 | |
| Recombinant DNA reagent | pInducer20 | Addgene | RRID:Addgene_44012 | |
| Recombinant DNA reagent | pInducer20-puro | this paper | | NeoR of pInducer20 (Addgene_44012) was replaced with PuroR |
| Recombinant DNA reagent | pFLAG-CMV2 | Sigma | Cat # E7033 | |
| Recombinant DNA reagent | pEGFP-C1 | Clontech | | |
| Peptide, recombinant protein | FLAG Peptide | Sigma | Cat # F3290 | |
| Peptide, recombinant protein | HA peptide | MBL Life science | Cat # 3320 | HA tagged Protein PURIFICATION KIT |
| Peptide, recombinant protein | recombinant human IL-1β | R and D Systems | Cat # 201-LB-005 | |
| Peptide, recombinant protein | recombinant mouse IL-1β | BioLegend | Cat # 575102 | |
| Peptide, recombinant protein | recombinant human IL-17A | BioLegend | Cat # 570502 | |
| Peptide, recombinant protein | recombinant human TNF | BioLegend | Cat # 570104 | |
| Commercial assay or kit | Dynabeads Protein G | Invitrogen | Cat # 10004D | |
| Commercial assay or kit | Lambda Protein Phosphatase | NEB | Cat # P0753S | |
| Commercial assay or kit | Signal Enhancer HIKARI | nacalai tesque | Cat # 02270-81 | |
| Commercial assay or kit | Immobilon Forte Western HRP Substrate | Millipore | Cat # WBLUF0500 | |
| Commercial assay or kit | TRIzol Reagent | Invitrogen | Cat # 15596018 | |
| Commercial assay or kit | RNA Clean and Concentrator-5 | Zymo Research | Cat # R1014 | |
| Commercial assay or kit | PowerUp SYBR Green Master Mix | Applied Biosystems | Cat # A25742 | |
| Commercial assay or kit | IL-6 Mouse Uncoated ELISA Kit | Invitrogen | Cat # 88-7064-88 | |
| Chemical compound, drug | DSP (dithiobis(succinimidyl propionate)) | TCI | Cat # D2473 | |
| Chemical compound, drug | Pam3CSK4 | InvivoGen | Cat # tlrl-pms | |
| Chemical compound, drug | poly I:C | cytiva | Cat # 27473201 | |
| Chemical compound, drug | LPS | InvivoGen | Cat # tlrl-smlps | |
| Chemical compound, drug | R848 | InvivoGen | Cat # tlrl-r848-5 | |
| Chemical compound, drug | CpG DNA | InvivoGen | Cat # tlrl-1668-1 | ODN 1668 |
| Chemical compound, drug | MG-132 | Sigma | Cat # 474790 | |
| Chemical compound, drug | Actinomycin D | Sigma | Cat # A9415 | |
| Chemical compound, drug | Leptomycin B | Sigma | Cat # L2913 | |

## Mice

*Zc3h12a*-deficient mice have been described previously (*Matsushita et al., 2009*). *Zc3h12a*[S513A/S513A] knock-in mice were generated using CRISPR/Cas9-mediated genome-editing technology as previously described (*Fujihara and Ikawa, 2014*). Briefly, a pair of complementary DNA oligos was

annealed and inserted into pX330 (Addgene plasmid # 42230) (*Cong et al., 2013*). The plasmid was injected together with the donor single strand oligo into fertilized eggs of C57BL/6J mice. Successful insertion was confirmed by direct sequencing.

All mice were grown under specific pathogen-free environments. All animal experiments were conducted in compliance with the guidelines of the Kyoto University animal experimentation committee (Approval number: MedKyo21057).

## Reagents

Recombinant cytokines, TLR ligands, and chemical compounds were listed in the key resources table.

## Cell culture

HeLa cells, HEK293T cells, RAW264.7 cells, and MEFs were maintained in DMEM (nacalai tesque) with 10% fetal bovine serum (FBS), 1% Penicillin/Streptomycin (nacalai tesque), and 100 μM 2-Mercaptoethanol (nacalai tesque). Mycoplasma contamination was routinely tested and found negative.

For the preparation of bone marrow-derived macrophages (BMDMs), bone marrow cells were cultured in RPMI-1640 (nacalai tesque) with 10% FBS, 1% Penicillin/Streptomycin, 100 μM 2-mercaptoethanol, and 20 ng/ml of macrophage colony-stimulating factor (M-CSF) (BioLegend) for 6 days.

For the preparation of thioglycolate-elicited peritoneal exudate cells (PECs), mice were intraperitoneally injected with 2 ml of 4% (w/v) Brewer's thioglycollate medium. 3.5 days after the injection, peritoneal macrophages were collected and cultured in RPMI-1640 with 10% FBS, 1% Penicillin/Streptomycin, and 100 μM 2-mercaptoethanol.

## Plasmids

For the expression of FLAG-tagged proteins, pFLAG-CMV2 (Sigma) was used as a backbone. For the expression of HA- or Myc-tagged proteins, the FLAG sequence of pFLAG-CMV2 was replaced by HA- or Myc-sequence. Mouse *Zc3h12a* cDNA was inserted into these vectors as previously described (*Matsushita et al., 2009*). The coding sequences of 14-3-3 and βTRCP were amplified by using cDNAs derived from HeLa cell as templates and inserted into vectors above using In-Fusion HD Cloning Kit (Takara Bio). For Myc-IRAK1 expression vector, coding sequence of IRAK1 derived from HA-IRAK1 expression vector (*Iwasaki et al., 2011*) was used. For the mouse *Il6* expression vector, the EGFP sequence in pEGFP-C1 was replaced with *Il6* gene.

Deletions or point mutations were introduced using the QuikChange Lightning Site-Directed Mutagenesis Kit (Agilent) or In-Fusion HD Cloning Kit.

For the lentiviral packaging vectors, pInducer20 (Addgene plasmid #44012) (*Meerbrey et al., 2011*) was modified to generate pInducer20-puro. FLAG-HA-Regnase-1 sequence was inserted into pInducer20-puro using In-Fusion HD Cloning Kit.

For the Cas9 and gRNA expression plasmids (pX459-IRAK1 and pX459-IRAK2), pX459 (Addgene Plasmid #62988) was digested at BbsI sites, and the annealed oligo coding guide sequence (key resources table) was inserted.

## Plasmid transfection

Plasmids were transfected to HeLa cells or HEK293T cells using Lipofectamine 2000 (Invitrogen) or PEI max (Polysciences) respectively according to manufacturer's instructions.

## Generation of doxycycline-inducible FLAG-HA-Regnase-1-expressing HeLa cells

HeLa cells expressing FLAG-HA-Regnase-1 in a doxycycline-dependent manner were generated by lentiviral transduction. To produce lentivirus, HEK293T cells were transfected with pInducer20-puro-FLAG-HA-Regnase1 together with third generation lentiviral packaging vectors. 6 hr after the transfection, the medium was changed to fresh medium and then the cells were incubated at 37℃ for 48 hr. After the incubation, the medium containing lentivirus was harvested and filtrated through 0.45 μm filter. HeLa cells were incubated with the virus-containing medium at 37℃ for 24 hr, followed by 48-hr-incubation with fresh medium. The transduced cells were selected by 0.5 μg/ml of puromycin

(InvivoGen). Single clones were picked and evaluated for their expression of FLAG-HA-Regnase-1 in a dox-dependent manner by immunoblotting.

## Knockout of *IRAK1* and *IRAK2*

HeLa cells were transfected with two pX459 plasmids which contains gRNA sequence for *IRAK1* and *IRAK2*. As the negative control, empty pX459 plasmid was transfected. Forty-eight hr after the pX459 transfection, puromycin (2 µg/ml) was added to the medium. After 48 hr selection with puromycin, the same number of cells were seeded to new dishes and incubated in fresh media without antibiotics for 48 hr. Knockout efficiency was check by immunoblotting using WCL samples.

## DSP-crosslinking

Doxycycline-inducible FLAG-HA-Regnase-1-expressing HeLa cells were treated with doxycycline (1 µg/ml, Sigma) and incubated at 37°C for 4 hr before the DSP-crosslinking. As a negative control, cells were incubated without doxycycline, and for the IL-1β-stimulated sample, cells were stimulated with human IL-1β (10 ng/ml, R and D Systems) 2 hr before the crosslinking. After the incubation, cells were washed twice with pre-warmed PBS, and then incubated in PBS containing 0.1 mM DSP (TCI) at 37°C for 30 min. After crosslinking, cells were washed once with pre-warmed PBS and incubated in STOP solution (PBS containing 1 M Tris-HCl pH 7.4) at room temperature for 15 min. Cells were then washed with ice-cold PBS twice, followed by cell lysis and immunoprecipitation.

## Immunoprecipitation

Before immunoprecipitation, pre-washed Dynabeads Protein G (Invitrogen) were incubated with either anti-FLAG antibody (Sigma), anti-HA antibody (Sigma), or anti-Myc antibody (Sigma) at 4°C with rotation for 1 hr.

For DSP-crosslinked samples, cells were lysed in IP buffer (20 mM Tris-HCl pH 7.4, 150 mM NaCl, and 0.5% (vol/vol) NP-40) with cOmplete Mini EDTA-free (Sigma), PhosSTOP (Sigma), and 200 U/ml of Benzonase (Millipore) and incubated on ice for 10 min. The lysates were centrifuged at 15,000 rpm for 5 min and the supernatants were incubated with anti-FLAG-antibody-bound Dynabeads at 4°C with rotation for 2 hr. The beads were then washed with IP buffer three times and incubated in FLAG-elution buffer (100 µg/ml FLAG peptides (Sigma), 50 mM Tris-HCl pH7.4, and 150 mM NaCl) at 4°C with rotation for 10 min twice. Eluted proteins were then immunoprecipitated using anti-HA-antibody-bound Dynabeads at 4°C with rotation for 2 hr. After the second immunoprecipitation, the beads were washed three times with IP buffer and the proteins were eluted in Urea elution buffer (8 M Urea and 50 mM Tris-HCl pH 8.0). The samples were stored at -80°C until trypsin digestion. Proteins were reduced with 10 mM dithiothreitol (Fujifilm Wako) for 30 min, alkylated with 50 mM iodoacetamide (Fujifilm Wako) for 30 min in the dark, diluted fourfold with 50 mM ammonium bicarbonate (ABC) buffer, and then trypsin digestion was performed. After overnight incubation, digestion was stopped by adding trifluoroacetic acid (TFA) (Fujifilm Wako) to a final concentration of 0.5%. The peptide mixture solution was desalted with SDB-XC StageTips (*Rappsilber et al., 2007*). The eluates were dried and resuspended in 200 mM 2-[4-2(2-hydroxyethyl)-1-piperazine]ethanesulfonic acid (HEPES) pH 8.5, mixed with 0.1 mg of TMT10-plex labeling reagents (Thermo Fisher Scientific) dissolved in 5 µL acetonitrile (ACN), and incubated for 1 hr at room temperature. The reaction mixtures were quenched by adding hydroxylamine (Sigma) to give a final concentration of 0.33%. After 15 min incubation, the samples were acidified with trifluoroacetic acid, diluted to 5% ACN, and desalted using SDB-XC StageTips. Peptides were dried, resolved in 5 mM ABC buffer and fractionated with a C18-StageTip. Peptides were eluted with 5 mM ABC containing acetonitrile (12.5%, 15%, 17.5%, 20%, 22.5% and 80%) in step gradient manner. Totally six fractions were obtained and analyzed by LC/MS/MS.

For the identification of phosphorylation sites of Regnase-1, HeLa cells expressing FLAG-HA-Regnase-1 or FLAG-Regnase-1 were stimulated with IL-1β (10 ng/ml) or IL-17A (50 ng/ml) respectively for 4 hr. The cells were washed with ice-cold PBS twice and lysed in IP buffer with cOmplete Mini EDTA-free and PhosSTOP. Regnase-1 was immunoprecipitated using anti FLAG antibody as described above and eluted from Dynabeads in SDS sample buffer (50 mM Tris–HCl pH 6.8, 2% (wt/vol) SDS, 15% (vol/vol) 2-mercaptoethanol, 10% (vol/vol) glycerol and bromophenol blue), followed by incubation at 95°C for 5 min. Regnase-1was isolated by electrophoresis and the pieces of the gel

containing Regnase-1 was stored at 4°C until trypsin digestion. The gels were de-stained for 30 min with 200 µL of 50 mM ABC / 50% ACN. Then the gels were dehydrated by the addition of 100% ACN. Proteins were reduced with 500 µL of 10 mM dithiothreitol / 50 mM ABC for 30 min, alkylated with 50 mM iodoacetamide / 50 mM ABC for 30 min in the dark. The gels were washed two times with 200 µL of 0.5% acetic acid / 50% methanol. After washing, gels were re-equilibrated with 50 mM ABC, and subsequently dehydrated by the addition of 100% ACN. 10 µL of trypsin solution (10 ng/µL in 50 mM ABC) was added to gel pieces and incubated for 5 min. Another 50 µL of 50 mM ABC buffer was added to gel samples and incubated at 37°C for overnight. After that, elastase (Promega) (150 ng/µL in water) was added to the final concentration of 7.5 ng/µL and incubated for 30 min at 37°C (*Dau et al., 2020*). Digestion was stopped by the addition of 5 µL of 10% TFA. The supernatants were recovered into fresh Eppendorf tubes, and two additional extraction steps were performed with 50% ACN / 0.1% TFA and 80% ACN / 0.1% TFA. The peptides in the supernatants were dried, resuspended in 0.1% TFA, and desalted using SDB-XC StageTips.

For detecting protein-protein binding, cells were lysed in IP Buffer with cOmplete Mini EDTA-free and PhosSTOP and immunoprecipitated as described above using indicated antibodies. The proteins were eluted in the mixture of IP Buffer and SDS sample buffer (2:1) and incubated at 95°C for 5 min.

For detecting protein-RNA binding, cells were lysed in IP Buffer with cOmplete Mini EDTA-free and RNaseOut (Invitrogen) and immunoprecipitated as described above using indicated antibodies. Some of the precipitates were eluted in the mixture of IP Buffer and SDS sample buffer (2:1) to elute proteins and the others were eluted in TRIzol Reagent (Invitrogen) for RNA isolation.

## LC-MS/MS

LC/MS/MS analyses were performed with an Orbitrap Fusion Lumos (Thermo Fisher Scientific) connected to an Ultimate 3000 pump (Thermo Fisher Scientific) and an HTC-PAL autosampler (CTC analytics). Peptides were separated by a self-pulled needle column (150 mm length, 100 µm ID, 6 µm needle opening) packed with Reprosil-Pur 120 C18-AQ 3 µm reversed-phase material (Dr. Maisch GmbH), using a 20 min or 65 min gradient of 5–40% B (mobile phase A: 0.5% acetic acid, mobile phase B: 0.5% acetic acid / 80% acetonitrile) at a flow rate of 500 nL/min. The applied ESI voltage was 2.4 kV. For TMT-labeled samples, the following parameters were applied: MS scan range of 375–1500, MS1 orbitrap resolution of 120,000, quadrupole isolation window of 0.7, HCD (higher-energy collision dissociation) collision energy of 38%, MS2 orbitrap resolution of 50,000, AGC target value of 50000. For non-labeled samples, the following parameters were applied: MS scan range of 300–1500, MS1 orbitrap resolution of 120,000, quadrupole isolation window of 1.6, HCD collision energy of 30%, MS2 orbitrap resolution of 15,000, MS2 AGC target value of 50,000.

## Database searching and data processing

For DSP-crosslinked samples, peptides were identified with Mascot version 2.6.1 (Matrix Science) against the sequence of Mouse Regnase-1 in addition to the human database from UniprotKB/Swiss-Prot release 2017/04 and with a precursor ion mass tolerance of 5 ppm and a product ion mass tolerance of 20 ppm. Carbamidomethyl (C), TMT6plex (K) and TMT6plex (N-term) were set as a fixed modification, oxidation (M) was allowed as a variable modification, and up to two missed cleavages are allowed with strict Trypsin/P specificity. Identified peptides were rejected if the Mascot score was below the 95% confidence limit based on the identity score of each peptide. The quantification of peptides was based on the TMT reporter ion intensities in MS2 spectra. Protein quantitative values were calculated by summing the corresponding peptide intensity values. Only proteins with at least two unique peptides were used for further analysis.

For the identification of phosphorylation sites of Regnase-1, peptides were identified with Mascot version 2.7.0 against the sequence of mouse Regnase-1 with a precursor ion mass tolerance of 5 ppm and a product ion mass tolerance of 20 ppm. Carbamidomethyl (C) was set as a fixed modification, oxidation (M) and phosphorylation (STY) were allowed as variable modifications, and up to two missed cleavages are allowed with semitrypsin specificity. Identified peptides were rejected if the Mascot score was below the 99% confidence limit based on the identity score of each peptide. The label-free quantification of peptides was based on the peak area in the extracted ion chromatograms using Skyline-daily software version 21.0.9.118 (*MacLean et al., 2010*). The peak area ratios

between stimulated and non-stimulated samples were calculated, log-scaled, and normalized by the median. For quantitation of phosphosites, the peak area ratios of all monophosphopeptides containing the phosphosites of interest were averaged. Phosphosite localization was evaluated with a site-determining ion combination method based on the presence of site-determining y- or b-ions in the peak lists of the fragment ions, which supported the phosphosites unambiguously (*Nakagami et al., 2010*).

Protein-protein interaction network of the Regnase-1-associating proteins (Log$_2$ fold change over negative control > 2) was analyzed using STRING database (*Szklarczyk et al., 2019*) and visualized in Cytoscape (*Shannon et al., 2003*). Keratins contaminated in the samples were omitted from the analysis.

## λ-protein phosphatase (λPP) treatment

HeLa cells transiently expressing HA-14-3-3ε were stimulated with or without IL-1β (10 ng/ml) for 4 hr and lysed in IP Buffer. Some of the lysates were used in immunoprecipitation as described above. The proteins were eluted using 250 μg/ml of HA peptides as described above. The lysate and the precipitates were treated with Lambda Protein Phosphatase (NEB) according to manufacturer's instructions. For the λPP-negative samples, the same amount of IP Buffer was added instead of λPP.

## Immunoblotting

Cells were lysed in IP Buffer or RIPA buffer (1% (vol/vol) NP-40, 0.1% (wt/vol) SDS, 1% (wt/vol) sodium deoxycholate, 150 mM NaCl, 20 mM Tris-HCl pH 8.0, and 10 mM EDTA) with cOmplete Mini EDTA-free and PhosSTOP. The lysates were incubated on ice for 5 min and centrifuged at 15,000 rpm for 5 min. The supernatants were mixed with SDS sample buffer (2:1) and incubated at 95°C for 5 min. SDS-PAGE was performed using e-PAGEL 7.5% or 5~20% (ATTO) and the proteins were transferred onto 0.2 μm pore size Immun-Blot PVDF membranes (Bio-Rad), followed by blocking with 5% skim milk. The antibodies used in immunoblotting were listed in the key resources table. Luminescence was detected with Amersham Imager 600 (cytiva) and the images were analyzed with Fiji (*Schindelin et al., 2012*).

## RNA isolation and RT-qPCR

Cells were lysed in TRIzol Reagent, and the RNA was isolated according to manufacturer's instructions. For the isolation of the RNA precipitated with Regnase-1, RNA was isolated using RNA Clean and Concentrator-5 (Zymo Research). RNA was reverse transcribed by using ReverTra Ace (TOYOBO). cDNA was amplified by using PowerUp SYBR Green Master Mix (Applied Biosystems) and measured with StepOnePlus Real-Time PCR System (Applied Biosystems). To analyze mRNA expression, each RNA level was normalized with 18S or ACTB. The primers used in qPCR were listed in *Supplementary file 1*.

## RNA sequencing

PECs were harvested from *Zc3h12a*[WT/WT] and *Zc3h12a*[S513A/S513A] mice as described above. PECs were stimulated with LPS (100 ng/ml) for indicated time and the RNA was collected and isolated using TRIzol Reagent. cDNA library was prepared using NEBNext Ultra RNA Library Prep Kit for Illumina (NEB) and sequenced on NextSeq 500 System (Illumina) according to the manufacturer's instructions. Acquired data was analyzed using Galaxy (*Afgan et al., 2018*). Briefly, identified reads were mapped on the murine genome (mm10) using HISAT2 (paired end, unstranded) (Galaxy Version 2.1.0), and the mapped reads were counted using featureCounts (Galaxy Version 1.6.3).

## ELISA

Cytokine concentration was measured by using IL-6 Mouse Uncoated ELISA Kit (Invitrogen) according to manufacturer's instructions. Luminescence was detected with iMark Microplate Reader (Bio-Rad).

## Luciferase assay

5xNF-κB firefly luciferase reporter vector, Renilla luciferase vector, and IRAK1-expressing vector were transfected in HeLa cells and the luciferase activity was measured by using PicaGene Dual Sea

Pansy Luminescence Kit (TOYO B-Net). NF-κB activation was calculated by normalizing Firefly luciferase activity with Renilla luciferase activity.

## Mathematical model

We developed two dynamical models for the inflammation system regulated by Regnase-1 based on different assumptions of the functions of 14-3-3-bound Regnase-1.

### Model 1

In the first model, we assumed that the 14-3-3-bound Regnase-1 does not have the function of degrading its target mRNAs (*Figure 4J*). The ordinary differential equations are given as follows:

$$\frac{dx_1}{dt} = k_1 signal(t) - d_1 x_1 x_3 - d_4 x_1 \tag{1.1}$$

$$\frac{dx_2}{dt} = k_2 signal(t) - d_2 x_2 x_3 - d_5 x_2$$

$$\frac{dx_3}{dt} = k_3 x_2 - (d_3 + d_6 signal(t) + d_7 signal(t))x_3 + d_9 x_4$$

$$\frac{dx_4}{dt} = d_7 signal(t)x_3 - (d_8 + d_9)x_4$$

where $x_1$, $x_2$, $x_3$, and $x_4$ is the abundance of *Il6* mRNA, *Zc3h12a* mRNA, Reg1 Protein, and 14-3-3-bound Reg1 Protein, respectively; $k_1$ and $k_2$ is the transcription rate constant of *Il6*, and *Zc3h12a*, respectively; $k_3$ is the translation rate constant of *Zc3h12a*; $d_1$ and $d_2$ is the Reg1-induced degradation rate constant of *Il6* mRNA and *Zc3h12a* mRNA, respectively; $d_3$, $d_4$, and $d_5$ is the Reg1-independent degradation rate constant of Reg1 protein, *Il6* mRNA, and *Zc3h12a* mRNA, respectively; $d_6$ is the ubiquitin-dependent degradation rate constant of Reg1 protein; $d_7$ is the binding rate constant of Reg1 protein to 14-3-3; $d_8$ is the natural degradation rate constant of 14-3-3-bound Reg1 protein; $d_9$ is the dissociation rate constant of Reg1 from 14-3-3. $signal(t)$ is the strength of TLR stimulation, which is given as the following form (*Mino et al., 2019*):

$$signal(t) = \begin{cases} s_{base} \left( if\ 0 \leq t \leq t_{delay} \right), \\ \frac{s_{input} - s_{base}}{t_{raise}} \left( t - t_{delay} \right) + s_{base} \left( if\ t_{delay} \leq t \leq t_{delay} + t_{raise} \right), \\ s_{input} \left( if\ t_{delay} + t_{raise} \leq t \leq t_{delay} + t_{raise} + t_{pulse} \right), \\ \left( s_{input} - s_{base} \right) \times \exp\left( -\frac{-t(t_{delay} + t_{raise} + t_{pulse})}{t_{delay}} \right) + sinput \\ \left( if\ t > t_{delay} + t_{raise} + t_{pulse} \right) \end{cases} \tag{1.2}$$

### Model 2

We also developed an alternative model in which the 14-3-3-bound Regnase-1 maintains functions of degrading its target mRNAs (*Figure 4J*). The ordinary differential equations are given as follows:

$$\frac{d\hat{x}_1}{dt} = k_1 signal(t) - d_1 \hat{x}_1 \hat{x}_3 - d_1' \hat{x}_1 \hat{x}_4 - d_4 \hat{x}_1 \tag{1.3}$$

$$\frac{d\hat{x}_2}{dt} = k_2 signal(t)d_2 - \hat{x}_2 \hat{x}_3 - d_2' \hat{x}_2 \hat{x}_4 - d_5 x_2$$

$$\frac{d\hat{x}_3}{dt} = k_3 x_2 - (d_3 + d_6 signal(t) + d_7 signal(t))\hat{x}_3 + d_9 \hat{x}_4$$

$$\frac{d\hat{x}_4}{dt} = d_7 signal(t)\hat{x}_3 - (d_8 + d_9)\hat{x}_4$$

where $\hat{x}_1$, $\hat{x}_2$, $\hat{x}_3$, and $\hat{x}_4$ is the abundance of *Il6* mRNA, *Zc3h12a* mRNA, Reg1 Protein, 14-3-3-bound Reg1 Protein, respectively; $d_1'$ and $d_2'$ is the 14-3-3-bound Reg1-induced degradation rate constant of *Il6* mRNA and *Zc3h12a* mRNA, respectively. The other parameters are defined in the same way as Model 1.

To determine which model is consistent with the experimental observations, we focus on the experimental findings that there was no difference in the abundance of *Il6* mRNA, *Zc3h12a* mRNA, and Reg1- protein (without 14-3-3 bound) between *Zc3h12a*^WT/WT^ and *Zc3h12a*^S513A/S513A^ cells in the late phase of stimulation (*Figure 4A,B,D and E*). We will show that in Model 2 (*Equation 1.3*), the abundance of the *Il6* mRNAs should be different between *Zc3h12a*^WT/WT^ and *Zc3h12a*^S513A/S513A^ cells under the condition that amount of the 14-3-3-free Reg1 protein is comparable between them.

## Analysis of the equilibrium

### Lemma 1

For *Zc3h12a*^WT/WT^ cells, there exists only one nonnegative (biologically meaningful) equilibrium of the system (*Equation 1.3*) if and only if $d_3 + d_6 s_{input} + d_7 s_{input} - \frac{d_7 d_9 s_{input}}{d_7 s_{input} + d_9} \geq 0$. If $d_3 + d_6 s_{input} + d_7 s_{input} - \frac{d_7 d_9 s_{input}}{d_7 s_{input} + d_9} < 0$, there is no equilibrium. For *Zc3h12a*^S513A/S513A^ cells, there always exists only one nonnegative (biologically meaningful) equilibrium.

### Proof of lemma 1

Setting all the derivatives of *Equation (1.3)* equal to zero yields

$$0 = k_1 s_{input} - d_1 \hat{X}_1^{WT} \hat{X}_3^{WT} - d_1' \hat{X}_1^{WT} \hat{X}_4^{WT} - d_4 \hat{X}_1^{WT} \tag{1.4}$$

$$0 = k_2 s_{input} - d_2 \hat{X}_2^{WT} \hat{X}_3^{WT} - d_2' \hat{X}_2^{WT} \hat{X}_4^{WT} - d_5 \hat{X}_2^{WT}$$

$$0 = k_3 \hat{X}_2^{WT} - \left(d_3 + d_6 s_{input} + d_7 s_{input}\right) \hat{X}_3^{WT} + d_9 \hat{X}_4^{WT}$$

$$0 = d_7 s_{input} \hat{X}_3^{WT} - (d_8 + d_9) \hat{X}_4^{WT}$$

where $\hat{X}_1^{WT}$, $\hat{X}_2^{WT}$, $\hat{X}_3^{WT}$, and $\hat{X}_4^{WT}$ are fixed points of $\hat{x}_1$, $\hat{x}_2$, $\hat{x}_3$, and $\hat{x}_4$, respectively. Given that $signal(t) \to s_{input}$ as $t \to \infty$, we assume $signal(t) \approx s_{input}$ at the equilibrium.

It follows from *Equation (1.4)* that

$$\left(d_2 + \frac{d_7 s_{input}}{d_8 + d_9} d_2'\right) K \left(\hat{X}_3^{WT}\right)^2 + d_5 K \hat{X}_3^{WT} - k_2 s_{input} = 0 \tag{1.5a}$$

$$\hat{X}_4^{WT} = \frac{d_7 s_{input}}{d_8 + d_9} \hat{X}_3^{WT} \tag{1.5b}$$

$$\hat{X}_2^{WT} = K \hat{X}_3^{WT} \tag{1.5c}$$

$$\hat{X}_1^{WT} = \frac{k_1 s_{input}}{d_1 \hat{X}_3^{WT} + d_1' \hat{X}_4^{WT} + d_4} \tag{1.5d}$$

where

$$K := \frac{1}{k_3}\left(d_3 + d_6 s_{input} + d_7 s_{input} - \frac{d_7 d_9 s_{input}}{d_7 s_{input} + d_9}\right)$$

It is easy to see that the quadratic *Equation (1.5a)* has a nonnegative solution if $K \geq 0$, that is, $d_3 + d_6 s_{input} + d_7 s_{input} - \frac{d_7 d_9 s_{input}}{d_7 s_{input} + d_9} \geq 0$. If $K < 0$, *Equation (1.5a)* has no nonnegative solution. If $\hat{X}_3^{WT} \geq 0$, it follows from (*Equation 1.5b*), (*Equation 1.5c*), and (*Equation 1.5d*) that $\hat{X}_4^{WT}$, $\hat{X}_2^{WT}$, $\hat{X}_1^{WT} \geq 0$.

For *Zc3h12a*^S513A/S513A cells, we assume that $d_7 = d_8 = d_9 = 0$. Substituting this equation into *Equation (1.4)* yields

$$0 = k_1 s_{input} - d_1 \hat{X}_1^{SA} \hat{X}_3^{SA} - d_4 \hat{X}_1^{SA} \tag{1.6}$$

$$0 = k_2 s_{input} - d_2 \hat{X}_2^{SA} \hat{X}_3^{SA} - d_5 \hat{X}_2^{SA}$$

$$0 = k_3 \hat{X}_2^{SA} - (d_3 + d_6 s_{input}) \hat{X}_3^{SA}$$

$$0 = \hat{X}_4^{SA}$$

where $\hat{X}_1^{SA}$, $\hat{X}_2^{SA}$, $\hat{X}_3^{SA}$, and $\hat{X}_4^{SA}$ are fixed points of $\hat{x}_1$, $\hat{x}_2$, $\hat{x}_3$, and $\hat{x}_4$ in *Zc3h12a*^S513A/S513A cells, respectively.

It follows from (1.6) that

$$d_2 \frac{k_3}{d_3 + d_6 s_{input}} \left(\hat{X}_2^{SA}\right)^2 + d_5 \hat{X}_2^{SA} - k_2 s_{input} = 0 \tag{1.7a}$$

$$\hat{X}_3^{SA} = \frac{k_3}{d_3 + d_6 s_{input}} \hat{X}_2^{SA} \tag{1.7b}$$

$$\hat{X}_1^{SA} = \frac{k_1 s_{input}}{d_1 \hat{X}_3^{SA} + d_4} \tag{1.7c}$$

It is easy to see that the quadratic *Equation (1.7a)* has a nonnegative solution. If $\hat{X}_2^{SA} \geq 0$, it follows from (*Equation 1.7b*) and (*Equation 1.7c*) that $\hat{X}_3^{SA} \, \hat{X}_1^{SA} \geq 0$.

Lemma 2. There exists only one nonnegative (biologically meaningful) equilibrium of the system (*Equation 1.1*) if and only if $d_3 + d_6 s_{input} + d_7 s_{input} - \frac{d_7 d_9 s_{input}}{d_7 s_{input} + d_9} \geq 0$. If $d_3 + d_6 s_{input} + d_7 s_{input} - \frac{d_7 d_9 s_{input}}{d_7 s_{input} + d_9} < 0$, there is no equilibrium. For *Zc3h12a*^S513A/S513A cells, there always exists only one nonnegative (biologically meaningful) equilibrium.

Proof of lemma2:

With $d_1' = d_2' = 0$ in lemma 1, we get the same conclusion.

## Consistency with the experiments

The experimental observation shows that there was no difference in the abundance of Reg1 protein between *Zc3h12a*^WT/WT and *Zc3h12a*^S513A/S513A cells at the late phase of stimulation (*Figure 4A–C*), which implies

$$\hat{X}_3^{WT} \approx \hat{X}_3^{SA} \tag{1.8}$$

based on the alternative model (1.3).

From (*Equation 1.4*) and (*Equation 1.6*), we get

$$\hat{X}_1^{WT} = \frac{k_1 s_{input}}{d_1 \hat{X}_3^{WT} + d_1' \hat{X}_4^{WT} + d_4} \tag{1.9a}$$

$$\hat{X}_2^{WT} = \frac{k_2 s_{input}}{d_2 \hat{X}_3^{WT} + d_2' \hat{X}_4^{WT} + d_5}$$

$$\hat{X}_1^{SA} = \frac{k_1 s_{input}}{d_1 \hat{X}_3^{SA} + d_4} \tag{1.9b}$$

$$\hat{X}_2^{SA} = \frac{k_2 s_{input}}{d_2 \hat{X}_3^{SA} + d_5}$$

By (*Equation 1.8*), (*Equation 1.9a*), and (*Equation 1.9b*), we obtain

$$\hat{X}_1^{WT} < \hat{X}_1^{SA} \tag{1.10a}$$

$$\hat{X}_2^{WT} < \hat{X}_2^{SA} \tag{1.10b}$$

*Equation 1.10a* and *Equation 1.10b* implies that in Model 2, the abundance of *Il6* and *Zc3h12a* mRNA in *Zc3h12a*^WT/WT cells should be smaller than that in *Zc3h12a*^S513A/S513A cells at the late phase under the condition that amount of the Reg1 protein is comparable (*Equation 1.8*) between these two cells. It contradicts experimental observation that the abundance of the *Il6* and *Zc3h12a* mRNAs did not differ between *Zc3h12a*^WT/WT and *Zc3h12a*^S513A/S513A cells (*Figure 4D–I*). Thus, Model 2 (*Equation 1.3*) is not consistent with the experimental findings.

In contrast, in Model 1 (*Equation 1.1*), we assume from experimental findings that

$$X_3^{WT} \approx X_3^{SA} \tag{1.11}$$

just like (*Equation 1.8*), where $X_3^{WT}$ is the fixed point of $x_3$ in *Zc3h12a*^WT/WT cells and $X_3^{SA}$ is the fixed point of $x_3$ in *Zc3h12a*^S513A/S513A cells based on the model (*Equation 1.1*).

By substituting $d_7 = d_8 = d_9 = 0$ into (*Equation 1.9a*) and (*Equation 1.9b*), we obtain

$$X_1^{WT} = \frac{k_1 s_{input}}{d_1 X_3^{WT} + d_4} \tag{1.12a}$$

$$X_2^{WT} = \frac{k_2 s_{input}}{d_2 X_3^{WT} + d_5}$$

$$X_1^{SA} = \frac{k_1 s_{input}}{d_1 X_3^{SA} + d_4} \tag{1.12b}$$

$$X_2^{SA} = \frac{k_2 s_{input}}{d_2 X_3^{SA} + d_5}$$

where $X_1^{WT}$ and $X_2^{WT}$ are fixed points of $x_1$ and $x_2$, respectively in *Zc3h12a*^WT/WT cells and $X_1^{SA}$ and $X_2^{SA}$ are fixed points of $x_1$ and $x_2$, respectively in *Zc3h12a*^S513A/S513A cells.

By (*Equation 1.11*), (*Equation 1.12a*), and (*Equation 1.12b*), we obtain

$$X_1^{WT} \approx X_1^{SA} \tag{1.13a}$$

$$X_2^{WT} \approx X_2^{SA} \tag{1.13b}$$

In this case, (*Equation 1.13a*) and (*Equation 1.13b*) are in agreement with the experimental facts that that the abundance of the target mRNAs did not differ between *Zc3h12a*^WT/WT and *Zc3h12a*^S513A/S513A cells.

These mathematical analyses indicate that Model 1 (*Equation 1.1*), but not Model 2 (*Equation 1.3*), is consistent with the experimental findings.

## Immunofluorescence

Cells were cultured on cover glass, fixed with 4%-Paraformaldehyde Phosphate Buffer Solution (nacalai tesque), and permeabilized with 0.5% (vol/vol) Triton X-100 (nacalai tesque) in PBS, followed by incubation in blocking solution (2% (vol/vol) goat serum (FUJIFILM Wako Pure Chemical) and 0.1% (wt/vol) gelatin in PBS). The antibodies used in Immunofluorescence were listed in the key resources table. DNA was stained with Hoechst 33342 (Invitrogen). Fluorescence was detected with TCS SPE (Leica). Acquired images were analyzed with Fiji (*Schindelin et al., 2012*).

## Amino acid sequence analysis

Amino acid sequence of each protein was obtained from NCBI. The results of T-coffee alignment (*Notredame et al., 2000*) were visualized by using Jalview (*Waterhouse et al., 2009*). Secondary structure was predicted by using PSIPRED 4.0 (*Buchan and Jones, 2019*; *Jones, 1999*). NES prediction was performed by using LocNES (*Xu et al., 2015*).

# Acknowledgements

We thank S Ogawa and N Kakiuchi in Kyoto university for performing RNA sequencing and providing *ZC3H12A*-KO HeLa cells, Y Okumoto for secretarial assistance and lab members for helpful discussion. This work was supported by Japan Society for the Promotion of Science (JSPS) KAKENHI [18H05278]; AMED-FORCE [JP20gm4010002] from Japan Agency for Medical Research and Development and the JSPS through Core-to-Core Program. KA was supported by 'Kibou Projects' Scholarship for doctoral Students in Immunology. TM was funded by JSPS KAKENHI (19H03488), Grant-in-Aid for Scientific Research on Innovative Areas 'Genome Science' (221S0002 and 16H06279), Takeda Science Foundation, the Uehara Memorial Foundation, Shimizu Foundation for Immunology and Neuroscience, Naito Foundation, Senri Life Science Foundation, Nakajima Foundation, and Mochida Memorial Foundation for Medical and Pharmaceutical Research.

# Additional information

### Funding

| Funder | Grant reference number | Author |
|---|---|---|
| Japan Agency for Medical Research and Development | JP20gm4010002 | Osamu Takeuchi |
| Japan Society for the Promotion of Science | Core-to-Core Program | Osamu Takeuchi |
| Japan Society for the Promotion of Science | 221S0002 | Takashi Mino |
| Japan Society for the Promotion of Science | 16H06279 | Takashi Mino |
| Japan Society for the Promotion of Science | 18H05278 | Osamu Takeuchi |
| Japan Society for the Promotion of Science | 19H03488 | Takashi Mino |
| Japan Agency for Medical Research and Development | 21ae0121030 | Osamu Takeuchi |

The funders had no role in study design, data collection and interpretation, or the decision to submit the work for publication.

### Author contributions

Kotaro Akaki, Conceptualization, Formal analysis, Validation, Investigation, Visualization, Methodology, Writing - original draft; Kosuke Ogata, Formal analysis, Investigation, Visualization, Methodology, Writing - original draft; Yuhei Yamauchi, Formal analysis, Visualization, Methodology, Writing - original draft; Noriki Iwai, Ka Man Tse, Investigation, Writing - review and editing; Fabian Hia, Formal analysis, Writing - review and editing; Atsushi Mochizuki, Yasushi Ishihama, Supervision, Writing - review and editing; Takashi Mino, Conceptualization, Supervision, Funding acquisition, Writing - review and editing; Osamu Takeuchi, Conceptualization, Supervision, Funding acquisition, Project administration, Writing - review and editing

### Author ORCIDs

Kotaro Akaki ⓘ https://orcid.org/0000-0003-0059-3291
Kosuke Ogata ⓘ http://orcid.org/0000-0002-0634-3990

Fabian Hia [ID] http://orcid.org/0000-0002-7209-4312
Yasushi Ishihama [ID] http://orcid.org/0000-0001-7714-203X
Takashi Mino [ID] http://orcid.org/0000-0002-9562-008X
Osamu Takeuchi [ID] https://orcid.org/0000-0002-1260-6232

### Ethics

Animal experimentation: All animal experiments were conducted in compliance with the guidelines of the Kyoto University animal experimentation committee. (Approval number: MedKyo21057).

### Decision letter and Author response

Decision letter https://doi.org/10.7554/eLife.71966.sa1
Author response https://doi.org/10.7554/eLife.71966.sa2

# Additional files

### Supplementary files

- Source data 1. Raw data of the results of immunoblotting are zipped in Source Data Files.
- Supplementary file 1. Primer sequences used in qPCR.
- Transparent reporting form

### Data availability

Mass spectrometry data (PXD026561) is available at https://repository.jpostdb.org/entry/JPST001201.

The following datasets were generated:

| Author(s) | Year | Dataset title | Dataset URL | Database and Identifier |
|---|---|---|---|---|
| Takeuchi O | 2021 | Phosphorylation-dependent Regnase-1 interactome | https://repository.jpostdb.org/entry/JPST001201 | JPOST, PXD026561 |
| Akaki K, Hia F, Kakiuchi N, Ogawa S, Takeuchi O | 2021 | Transcriptome analysis of LPS stimulated PECs (WT and Zc3h12a S513A) | https://www.ncbi.nlm.nih.gov/geo/query/acc.cgi?acc=GSE180028 | NCBI Gene Expression Omnibus, GSE180028 |

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
