## [Decision Letter]

**Acceptance summary:**

This study describes a new mechanism by which Regnase-1 is inhibited upon immune activation, which mediates the efficient synthesis of inflammatory mediators whose mRNAs are normally degraded by Regnase-1. The study report a novel interaction of Regnase-1 with 14-3-3. This provides an alternative mechanism by which inflammatory mRNAs are upregulated by inhibiting degradation via Regnase-1.

**Decision letter after peer review:**

Thank you for submitting your article "IRAK1-dependent Regnase-1-14-3-3 complex formation controls Regnase-1-mediated mRNA decay" for consideration by *eLife*. Your article has been reviewed by 3 peer reviewers, and the evaluation has been overseen by a Reviewing Editor and Carla Rothlin as the Senior Editor. The following individual involved in the review of your submission has agreed to reveal their identity: Georg Stoecklin (Reviewer #2).

Essential revisions:

Most of the reviewers' excellent suggestions are intended for future experiments, which are not required for this paper. However, some experiments lack necessary controls and others do not adequately support the conclusions. The comments listed before must be addressed, and some statements (see specific reviewers' comments) need to be changed or deleted:

1) Figure 1C. Input control on the same blots as the IPs are run ought to be shown, and include a negative control blot (e.g. actin) as well.

2) Figure 1D. It is important to show a longer exposure of the Regnase1 blots for both the input blot and the blot from the immunoprecipitation.

3) You show that IL-6 mRNA levels are similar in Regnase-1 WT and S513A mutant cells (Figure 4D-F). This notion is difficult to reconcile with the author's finding that 14-3-3 binding to Regnase-1 prevents it from binding to and inducing the degradation of IL-6 mRNA. This discrepancy should be discussed critically. Moreover, previous studies have shown that changes in mRNA stability are not necessarily reflected by changes in mRNA steady-state levels, since they can be buffered by altered transcription (see e.g. Singh et al., 2019; PMID: 31116665). Therefore, mRNA degradation rates should be measured directly by actinomycin D chase or similar experiments.

4) You claim "that 14-3-3 inhibits Regnase-1-mRNA binding, thereby abrogating Regnase-1-mediated mRNA degradation". However, this is only shown for IL-6 mRNA (Figure 5G). Since this is a key result of the study, you should also test other targets of Regnase-1 so as to generalize their finding.

5) Figure 5A. Expression levels of exogenous Regnase-1 mutants ought to be shown.

*Reviewer #2 (Recommendations for the authors):*

The data suggest that IRAK1/2 may directly phosphorylate Regnase-1 (Fig.2G-I), although the authors do not address this question either experimentally or in the discussion. Do the authors have evidence that Regnase-1 is a direct target of IRAK1/2? Minimally, the authors should discuss this point and assess whether the identified phosphorylation sites conform to consensus IRAK target motifs.

The evidence for mutually exclusive binding of Regnase-1 to bTRCP or 14-3-3 is rather indirect, through the analysis of Regnase-1 phosphorylation status and phosphomutants (Fig.3). This point could be strengthened by competition assays, in which expressing increasing amounts of one protein should weaken the interaction with the other.

Figure 2G: The figure legend states that HeLa cells were "stimulated with IL-1b (10 ng/ml) for 4 hours". Is this correct? The blot shows the faster migrating Regnase-1 band and no interaction with 14-3-3 unless IRAK1/2 is overexpressed, which corresponds to the situation in unstimulated cells.

Figure 4A-C: As a direct assessment of Regnase-1 protein stability, the authors could conduct cycloheximide chase assays in Wt/Wt and S513A/S513A cells.

Figure 4 and 5: The figure legends say "…bars represent mean values of biological replicates (n=3)… similar results were obtained from at least two independent experiments". It is unclear what is meant by these two different statements.

lines 189 and 195: should refer to Figure 3, not 2.

line 231: should refer to Figure 4D-I

Figure 4J-K: figure legend should say Regnase-1 instead of Reganse-1

*Reviewer #3 (Recommendations for the authors):*

Line 26: "both interactions occur in a mutually exclusive manner, underscoring the importance of modulating Regnase-1's activity"

I don't think the fact that the interactions are mutually exclusive underscores the importance of modulating Regnase activity. That Regnase-1 can be inactivated by two different, and perhaps partially redundant, mechanisms underscores the importance of modulating Regnase activity.

Line 138: "14-3-3 specifically binds to phosphorylated Regnase-1"

The authors concluded that 14-3-3 binds to phosphorylated S494 and S513 of Regnase-1 but direct binding between Regnase-1 and 14-3-3 and their binding sites are not shown. The authors would need to perform experiments to address this point, or otherwise rewrite the manuscript to correctly interpret the data.

In Figure 2A, the IL-1β-induced interaction between 14-3-3 and Reg1 is reduced by phosphatase treatment, but it still appears relatively strong. Therefore, it is not possible to say from this IP that 14-3-3 binds specifically to phospho-Reg1. Since the band shift is abolished by PP in Figure 2A, it looks like 14-3-3 may still bind to non-phosphorylated Reg1, but perhaps with lower affinity.

Line 190-191: "We next checked the phosphorylation status of βTRCP-bound and 14-3-3-bound Regnase-1"

This sentence seems to be in the wrong place. It appears to refer to Figure 3A/B, but it comes before 3C.

Line 195: "βTRCP-ΔF-bound Regnase-1 migrated faster than 14-3-3-bound Regnase-1 (Figure 2C-D)"

Please clarify why Figure2C and 2D are referenced here. In particular, Figure 2C shows residues which are phosphorylated in IL-1B-stimulated cells.

Line 211: "Regnase-1 showed slow migration and was able to associate with 14-3-3"

Based on the band shift in Figure 4A, it is a reasonable assumption that the newly synthesized Reg1 was phosphorylated and therefore it could bind to 14-3-3, but the figure does not show that it was associated with 14-3-3. This should be made clear.

Figure 5A, Expression levels of exogenous Regnase-1 mutants need to be shown. It is considered that the expression levels must be the same at initial condition.

Line 231: "Figure 3D-I" should refer to "Figure 4D-I"

Line 248: "These results suggest that the phosphorylation at S513 and the following association with 14-3-3 nullifies Regnase-1's ability in degrading target mRNAs"

From the data presented up to this point in the manuscript, it is not possible to say whether 14-3-3 binding or S513 phosphorylation alone prevents Reg1 from degrading Il6 mRNA. The following data presented in Figure 5 does address this.

Line 262: The experiments using Reg-1ExoSx2 are valid only if interacting site of 14-3-3 with Regnase-1 is the same as that with ExoS. The authors should clearly mention this point or show that 14-3-3 binds to Regnase-1 with its phosphopeptide binding groove.

Line 278: In Figure 5D, the difference between Il6 only and Il6+ExoSx2 looks like it could be significant. What is the P value?

Line 335-336: "Analysis of Regnase-1S513A/S513A mice revealed that 14-3-3-mediated abrogation of Regnase-1 can be compensated by the degradation of Regnase-1"

The data from the S513A mutants doesn't show that degradation compensates for 14-3-3-mediated block of Reg1 function. It shows that not all Reg1 is bound to 14-3-3, and only the non-14-3-3-bound Reg1 regulates mRNA stability.

The authors could say that the absence of 14-3-3-mediated abrogation of Reg1 in the S513A mutant cells may be compensated for by increased degradation. But to be sure of this, the authors should demonstrate that translation is not affected.

Line 347: "compensation by βTRCP-mediated regulation, which we observed in Regnase-1S513A/S513A mice"

Could the authors demonstrate that this is definitely βTRCP-mediated regulation? If the mutant βTRCP used in Figure 2 acts as a dominant-negative, perhaps overexpression in S513A mutant cells could show this?

Line 369-370: "βTRCP is likely to recognize 14-3-3-free Regnase-1, indicating that 14-3-3 inhibits Regnase-1-βTRCP interaction"

The wording of this sentence suggests that 14-3-3-Reg1 binding takes precedence over βTRCP-Reg1. Is there any reason to assume that this is true? If binding is mutually exclusive, as suggested by Figure 3C, could βTRCP also inhibit the 14-3-3-Reg1 interaction? The authors should clarify here whether the interaction of Reg1 with 14-3-3 is likely to be dominant over the interaction with βTRCP.

Figure 1C suggests that βTRCP-Reg1 binding peaks sooner after IL-1β stimulation than 14-3-3-Reg1 binding. Could this mean that βTRCP binds to Reg1 first, before 14-3-3?

Line 387: "evolutionally conserved" should be "evolutionarily conserved"

*Reviewer #4 (Recommendations for the authors):*

1. In Figure 1C, the co-immunoprecipitation Regnase1 with myc-tagged 14-3-3epsilon doesn't look exceptionally robust, especially when seeing the original blot provided as source data. It's currently hard to tell whether this represents a small amount of endogenous Regnase1 or a larger percentage co-precipitating. What would help to improve the strength of this figure is to include an input control on the same blots as the IPs are run, and to include a negative control blot (e.g. actin) as well. This will help to validate the specificity of the interaction.

2. For Figure 1D, are the authors able to show a longer exposure of the Regnase1 blots as well? For both the input blot and the blot from the immunoprecipitation. This would help to strengthen their conclusion that R848 and CpG DNA stimuli promote the interaction between 14-3-3 and Regnase1.

3. The authors conclude that the interaction of Regnase1 and 14-3-3 is mediated by IRAK-dependent phosphorylation. As these experiments were all conducted by overexpression of IRAK1 or 2, it would significantly strengthen their conclusion if they carried out some knockdown experiments where they deplete endogenous IRAK1/2. Does this abolish the 14-3-3 interaction with Regnase1 in IL-1B treated cells?

4. While I agree with the authors that their data support different phosphorylated serine residues supporting interactions with either 14-3-3 or TRCP, I don't think that their data specifically support mutually exclusive interactions between the two. A convincing competition experiment to test this may be to express both 14-3-3 and TRCP at different levels along with Regnase1. This would allow them to immunopreciptate a tagged Regnase 1 and see if, for example, increasing TRCP expression increases its association with Regnase1, while decreasing the association of 14-3-3 with Regnase1.

5. For Figure 5G, the authors should also test specific Serine-to-Alanine mutants to see if they impact RNA binding. This could be accomplished with a mutant that cannot bind TRCP (to keep it stable), and compare it to a mutant that cannot bind both TRCP and 14-3-3. This would allow the authors to make conclusions on a Regnase1 mutant that did not contain additional bacterial stretches to it, which could bind to proteins other than 14-3-3 that they don't know about.

6. Similarly, the final conclusion of the manuscript is that 14-3-3 binding to Regnase1 localizes to the nucleus. However, this is based on Regnase mutants that utilize bacterial stretches that bind 14-3-3 proteins (and potentially other proteins as well). To strengthen their data, the authors should utilize a Regnase mutant that cannot bind to TRCP and/or contains mutations of serine residues such that it cannot bind 14-3-3 proteins. These mutants could then be used to test if nuclear localization is impaired without adding additional bacterial protein stretches to it.

---

## [Author Response]

Essential revisions:Most of the reviewers' excellent suggestions are intended for future experiments, which are not required for this paper. However, some experiments lack necessary controls and others do not adequately support the conclusions. The comments listed before must be addressed, and some statements (see specific reviewers' comments) need to be changed or deleted:

We thank the reviewers and editors for valuable suggestions. We revised the manuscript accordingly as indicated in our responses to the reviewers’ comments below.

1) Figure 1C. Input control on the same blots as the IPs are run ought to be shown, and include a negative control blot (e.g. actin) as well.

We thank the reviewer for the advice. According to the reviewer’s suggestion, we loaded the input and IP samples used in Figure 1C on the same gel and blotted membranes using anti-Actin antibody as well as anti-Regnase-1 antibody (new Figure 1—figure supplement 1). Compared with input samples, only trace amounts of Actin bands were detected in every IP-sample including 14-3-3ε negative. This is probably because of non-specific binding between Actin and Dynabeads or protein G used for immunoprecipitation. In contrast, strong Regnase-1 bands were detected in IL-1β-stimulated (2 or 4 hours) HA-14-3-3ε transfected samples, but not in IL-1β-stimulated (4 hours) HA-14-3-3ε negative sample. Therefore, these results further support our finding that the binding between Regnase-1 and 14-3-3 specifically induced by IL-1β stimulation.

2) Figure 1D. It is important to show a longer exposure of the Regnase1 blots for both the input blot and the blot from the immunoprecipitation.

We thank the reviewer for the constructive criticism. Since we acquired the image with only one exposure time setting (16-bit), we adjusted the range of brightness to show the bands at two different brightness level (8-bit) (new Figure 1D). These data further confirm that the MyD88-dependent signaling activated downstream of TLR/IL-1R induces the interaction between Regnase-1 and 14-3-3.

3) You show that IL-6 mRNA levels are similar in Regnase-1 WT and S513A mutant cells (Figure 4D-F). This notion is difficult to reconcile with the author's finding that 14-3-3 binding to Regnase-1 prevents it from binding to and inducing the degradation of IL-6 mRNA. This discrepancy should be discussed critically. Moreover, previous studies have shown that changes in mRNA stability are not necessarily reflected by changes in mRNA steady-state levels, since they can be buffered by altered transcription (see e.g. Singh et al., 2019; PMID: 31116665). Therefore, mRNA degradation rates should be measured directly by actinomycin D chase or similar experiments.

We thank the reviewer for the suggestion. To check mRNA degradation rates, we stimulated MEFs with IL-1β for 4 hours, at which time point the protein level of Regnase-1 shows huge difference between *Regnase-1*^WT/WT^ and *Regnase-1*^S513A/S513A^ cells, and added Actinomycin D. We found that degradation rates of Regnase-1 target mRNAs such as *Il6*, *Regnase-1*, and *Nfkbiz* were comparable *Regnase-1*^WT/WT^ and *Regnase-1*^S513A/S513A^ MEFs (now shown as new Figure 4—figure supplement 4).

4). You claim "that 14-3-3 inhibits Regnase-1-mRNA binding, thereby abrogating Regnase-1-mediated mRNA degradation". However, this is only shown for IL-6 mRNA (Figure 5G). Since this is a key result of the study, you should also test other targets of Regnase-1 so as to generalize their finding.

We thank the reviewer for the constructive criticism. Since we previously identified binding targets of Regnase-1 in HeLa cells (Mino et al., 2015), we checked if ExoSx2 mutation also abrogates the binding between Regnase-1 and some of these target mRNAs including *PTGS2*, *NFKBIZ*, *REGNASE-1*, *CXCL1*, *CXCL2*, *CXCL3*, *MAFA*, and *NFKBID*. As shown in new Figure 5—figure supplement 2, Regnase-1-ExoSx2 failed to bind with all Regnase-1 target mRNAs tested, in addition to *IL6* shown in Figure 5H. These results further confirm that 14-3-3 inhibits Regnase-1-mRNA binding.

5). Figure 5A. Expression levels of exogenous Regnase-1 mutants ought to be shown.

We thank the reviewer for the comment. Since the majority of overexpressed Regnase-1 are not affected by IL-1β-mediated phosphorylation (e.g., Figure 2D, 3A, and 3B), we reduced the amount of transfected Regnase-1 as much as possible in former Figure 5A (Figure 5B) in order to observe the effect of each SA mutation on the activity of Regnase-1. However, such low expressed Regnase-1 was undetectable in immunoblotting with tag antibody. Therefore, to check the expression level of low expressed Regnase-1, we performed immunoblot analysis using anti-Regnase-1 antibody, which has strong affinity to Regnase-1, using *REGNASE-1* KO HeLa cells transfected with Regnase-1-WT and SA mutants (new Figure 5A). The expression level of each exogenous Regnase-1 was comparable without IL-1β stimulation. Nevertheless, the S513A mutant Regnase-1 showed decreased expression 2 or 4 hours after IL-1β stimulation, consistent with our observation in *Regnase-1*^S513A/S513A^ MEFs (Figure 4A). On the other hand, S435/439/513A mutant showed higher expression compared to S513A mutant.

Reviewer #2 (Recommendations for the authors):The data suggest that IRAK1/2 may directly phosphorylate Regnase-1 (Fig.2G-I), although the authors do not address this question either experimentally or in the discussion. Do the authors have evidence that Regnase-1 is a direct target of IRAK1/2? Minimally, the authors should discuss this point and assess whether the identified phosphorylation sites conform to consensus IRAK target motifs.

We thank the reviewer for the suggestion. A previous kinome study comprehensively identifying kinase substrates suggests that the sequence motifs of target phosphorylation site of IRAK1 is pSxV and KxxxpS (Sugiyama et al., 2019; PMID: 31324866). However, these sequences do not match the sequence at S494 and S513 of Regnase-1 (Figure 2F). We speculate other kinases are activated by IRAK1/2 and phosphorylate Regnase-1 at S494 and S513, although we cannot exclude the possibility that Regnase-1 is directly phosphorylated by IRAK1/2 at S494 and S513. We added this point in the Discussion section.

The evidence for mutually exclusive binding of Regnase-1 to bTRCP or 14-3-3 is rather indirect, through the analysis of Regnase-1 phosphorylation status and phosphomutants (Fig.3). This point could be strengthened by competition assays, in which expressing increasing amounts of one protein should weaken the interaction with the other.

We thank the reviewer for the criticism. As the other reviewers point out, the wording of "mutually exclusive" might mislead readers. As shown in Figure 3C and 3D, βTRCP recognizes 14-3-3-free Regnase-1 but not slowly migrating Regnase-1, which is the binding target of 14-3-3. In addition, Regnase-1-S513A mutant is unstable after IL-1β or LPS stimulation (Figure 4A, 4B, 4C, and 5A). These results suggest that 14-3-3 stabilizes Regnase-1 by preventing the formation of Regnase-1-βTRCP complex. However, we have not shown the data indicating βTRCP inhibits Regnase-1-14-3-3 association. We therefore corrected the sentences about the relationship between Regnase-1-14-3-3 complex and Regnase-1-βTRCP complex throughout the manuscript. These binding events occur independently and not sequentially, and 14-3-3 inhibits Regnase-1-βTRCP binding. We did not investigate whether βTRCP affects Regnase-1-14-3-3 interaction or not because once proteins (substrates of SCF complex) bind to βTRCP, they get ubiquitinated and degraded via proteasome system.We agree with the reviewer that the competition assay will help to clarify the detailed mechanism how 14-3-3 inhibits Regnase-1-βTRCP binding. However, we feel thein vitroassay is beyond the scope of this study. 14-3-3 mediated abrogation of the nuclear-cytoplasmic shuttling of Regnase-1 might be one of clues to answer this question.

Figure 2G: The figure legend states that HeLa cells were "stimulated with IL-1b (10 ng/ml) for 4 hours". Is this correct? The blot shows the faster migrating Regnase-1 band and no interaction with 14-3-3 unless IRAK1/2 is overexpressed, which corresponds to the situation in unstimulated cells.

We thank the reviewer for pointing out our mistake in the Figure legend. We did not stimulate cells in the experiment shown in Figure 2G. We corrected the legend for Figure 2G as follows: “(G) Immunoblot analysis of immunoprecipitates (IP: Myc) and WCL from HeLa cells transiently expressing Myc-14-3-3ε, HA-IRAK1/2, and FLAG-Regnase-1-WT or indicated mutants.”

Figure 4A-C: As a direct assessment of Regnase-1 protein stability, the authors could conduct cycloheximide chase assays in Wt/Wt and S513A/S513A cells.

We thank the reviewer for the advice. According to the reviewer’s advice, we assessed Regnase-1 protein stability by adding cycloheximide (CHX) (new Figure 4—figure supplement 2). Since both WT and S513A Regnase-1 degraded rapidly (30 minutes) after the stimulation with IL-1β or LPS (Figure 4A-C), we added CHX and LPS at the same time and collected cell lysates at earlier time points (10, 20, and 30 minutes after the stimulation). We found that Regnase-1 S513A mutant degraded more quickly than WT Regnase-1 in PECs in response to LPS stimulation (Figure 4—figure supplement 2). This is because phosphorylated WT Regnase-1 (upper bands), which was not observed in *Regnase-1*^S513A/S513A^, persisted after the stimulation. This result further supports our finding that S513A mutation destabilizes Regnase-1 protein.

Reviewer #3 (Recommendations for the authors):Line 26: "both interactions occur in a mutually exclusive manner, underscoring the importance of modulating Regnase-1's activity"I don't think the fact that the interactions are mutually exclusive underscores the importance of modulating Regnase activity. That Regnase-1 can be inactivated by two different, and perhaps partially redundant, mechanisms underscores the importance of modulating Regnase activity.

We thank the reviewer for the criticism. As the reviewer pointed out, the abstract lacks logic to some extent because of the order of sentences. According to the reviewer’s comment, we revised the abstract section.

Line 138: "14-3-3 specifically binds to phosphorylated Regnase-1"The authors concluded that 14-3-3 binds to phosphorylated S494 and S513 of Regnase-1 but direct binding between Regnase-1 and 14-3-3 and their binding sites are not shown. The authors would need to perform experiments to address this point, or otherwise rewrite the manuscript to correctly interpret the data.In Figure 2A, the IL-1β-induced interaction between 14-3-3 and Reg1 is reduced by phosphatase treatment, but it still appears relatively strong. Therefore, it is not possible to say from this IP that 14-3-3 binds specifically to phospho-Reg1. Since the band shift is abolished by PP in Figure 2A, it looks like 14-3-3 may still bind to non-phosphorylated Reg1, but perhaps with lower affinity.

We thank the reviewer for the advice. 14-3-3 is well known to directly bind to phosphorylated proteins and the binding motif of 14-3-3 partially overlaps with the sequences surrounding S494 and S513 of Regnase-1 (Figure 2F). Furthermore, the mutations in S494 and S513 abrogated the interaction between Regnase-1 and 14-3-3 (Figure 2D), indicating that the phosphorylation of both sites is critical for Regnase-1-14-3-3 interaction. These results strongly suggest that the 14-3-3 dimer directly recognizes phosphorylated Regnase-1 at both S494 and S513 at least under IL-1β-, MyD88-dependent TLR ligand-, or IL-17A-stimulation. We added this argument in the Discussion section.

In Figure 2A-B, we treated the samples with λPP after the immunoprecipitation with HA-14-3-3ε, indicating that Regnase-1 is supposed to be detected in the precipitated samples irrespective of the λPP treatment. Indeed, the amounts of Regnase-1 co-precipitated with 14-3-3ε was not altered by the λPP treatment (Figure 2A). However, the λPP treatment changed the mobility of Regnase-1 in the immunoblot analysis and made Regnase-1 migrate faster. These results demonstrate that 14-3-3 associates phosphorylated, but not unphosphorylated, Regnase-1.

Line 190-191: "We next checked the phosphorylation status of βTRCP-bound and 14-3-3-bound Regnase-1"This sentence seems to be in the wrong place. It appears to refer to Figure 3A/B, but it comes before 3C.

We thank the reviewer for the advice. According to the reviewer’s suggestion, we revised the sentence as follows: "We next compared the phosphorylation status of βTRCP-bound and 14-3-3-bound Regnase-1."

Line 211: "Regnase-1 showed slow migration and was able to associate with 14-3-3"Based on the band shift in Figure 4A, it is a reasonable assumption that the newly synthesized Reg1 was phosphorylated and therefore it could bind to 14-3-3, but the figure does not show that it was associated with 14-3-3. This should be made clear.

We thank the reviewer for the concern. To clarify the point raised by the reviewer, we revised the sentences as follows: " Notably, most of the newly synthesized Regnase-1 showed slow migration, consistent with the immunoprecipitation experiment using HeLa cells or RAW264.7 cells shown in Figure 1C and 1D… Interestingly, the amount of Regnase-1 at lower bands, which are not the binding target of 14-3-3 (Figure 2A), was comparable between WT and *Regnase-1*^S513A/S513A^ at corresponding time points."

Figure 5A, Expression levels of exogenous Regnase-1 mutants need to be shown. It is considered that the expression levels must be the same at initial condition.

This comment was found in the Essential Revisions and was already addressed.

Line 248: "These results suggest that the phosphorylation at S513 and the following association with 14-3-3 nullifies Regnase-1's ability in degrading target mRNAs"From the data presented up to this point in the manuscript, it is not possible to say whether 14-3-3 binding or S513 phosphorylation alone prevents Reg1 from degrading Il6 mRNA. The following data presented in Figure 5 does address this.

We thank the reviewer for the comment. We agree with the concern and corrected the sentences as follows:

"These results imply that the phosphorylation at S513 and the following association with 14-3-3 nullifies Regnase-1’s ability in degrading target mRNAs"

Line 262: The experiments using Reg-1ExoSx2 are valid only if interacting site of 14-3-3 with Regnase-1 is the same as that with ExoS. The authors should clearly mention this point or show that 14-3-3 binds to Regnase-1 with its phosphopeptide binding groove.

We thank the reviewer for the concern. As we mentioned above, it is quite plausible that the 14-3-3 dimer directly recognizes phosphorylated S494 and S513 of Regnase-1. In this regard, we believe that the experiments using Regnase-1-ExoSx2 are meaningful.

Line 278: In Figure 5D, the difference between Il6 only and Il6+ExoSx2 looks like it could be significant. What is the P value?

The P value between Il6 only and Il6+ExoSx2 is 0.00048. Since we overexpressed Regnase-1-ExoSx2, we speculate this difference is because some of 14-3-3-free ExoSx2 mutant functioned to suppress Il6 expression.

Line 335-336: "Analysis of Regnase-1S513A/S513A mice revealed that 14-3-3-mediated abrogation of Regnase-1 can be compensated by the degradation of Regnase-1"The data from the S513A mutants doesn't show that degradation compensates for 14-3-3-mediated block of Reg1 function. It shows that not all Reg1 is bound to 14-3-3, and only the non-14-3-3-bound Reg1 regulates mRNA stability.The authors could say that the absence of 14-3-3-mediated abrogation of Reg1 in the S513A mutant cells may be compensated for by increased degradation. But to be sure of this, the authors should demonstrate that translation is not affected.

We thank the reviewer for the concern. Reviewer #2 also suggests that we needed to make sure if S513A mutation affects protein stability of Regnase-1. According to the reviewers’ suggestion, we examined the stability of Regnase-1 by treating the cells with CHX. As shown in new Figure 4—figure supplement 2, we successfully show that Regnase-1-S513A is more unstable than WT after LPS and CHX treatment.

Line 347: "compensation by βTRCP-mediated regulation, which we observed in Regnase-1S513A/S513A mice"Could the authors demonstrate that this is definitely βTRCP-mediated regulation? If the mutant βTRCP used in Figure 2 acts as a dominant-negative, perhaps overexpression in S513A mutant cells could show this?

We thank the reviewer for the concern. As shown in new Figure 5A, additional S435/439A mutation, which abrogates Regnase-1-βTRCP binding, stabilized Regnase-1-S513A protein after IL-1β stimulation. The result indicates that the degradation of Regnase-1-S513A in response to IL-1β stimulation is dependent on βTRCP. Therefore, we revised the sentence as follows: "…compensation by βTRCP-mediated regulation, which we observed in Regnase-1S513A/S513A mice and HeLa cells transiently expressing Regnase-1-S513A and S435/439/513A."

Line 369-370: "βTRCP is likely to recognize 14-3-3-free Regnase-1, indicating that 14-3-3 inhibits Regnase-1-βTRCP interaction"The wording of this sentence suggests that 14-3-3-Reg1 binding takes precedence over βTRCP-Reg1. Is there any reason to assume that this is true? If binding is mutually exclusive, as suggested by Figure 3C, could βTRCP also inhibit the 14-3-3-Reg1 interaction? The authors should clarify here whether the interaction of Reg1 with 14-3-3 is likely to be dominant over the interaction with βTRCP.Figure 1C suggests that βTRCP-Reg1 binding peaks sooner after IL-1β stimulation than 14-3-3-Reg1 binding. Could this mean that βTRCP binds to Reg1 first, before 14-3-3?

We thank the reviewer for the concern. As the reviewer pointed out, both Regnase-1-WT and S513A undergo βTRCP-mediated degradation within 30 minutes after IL-1β or LPS stimulation (Figure 4A-C) (Iwasaki et al., 2011). This is probably because of the higher activity of IKKs at the early time point compared to later time points (Iwasaki et al., 2011). We think that Regnase-1 can also associate with 14-3-3 at such an early time point because Regnase-1-WT showed strong phosphorylated band 30 minutes after LPS stimulation when protein degradation was inhibited by MG-132 (Figure 4C). Moreover, we found that Regnase-1 phosphorylation, which was not observed in *Regnase-1*^S513A/S513A^, occurred 10 minutes after LPS stimulation (new Figure 4—figure supplement 2). These data indicate that Regnase-1 can be phosphorylated at S513 at the early time point after IL-1β or LPS stimulation. Since 14-3-3 does not covalently bind to Regnase-1, it is plausible that Regnase-1-14-3-3 binding is a reversible reaction, while protein degradation of Regnase-1 is irreversible. We believe this is the reason why 14-3-3 failed to protect Regnase-1 from βTRCP-mediated protein degradation at the early time point.

Reviewer #2 also pointed out the wording, "mutually exclusive", might not appropriate. As we described in our response to the comment of Reviewer #2, we have corrected the wording.

Reviewer #4 (Recommendations for the authors):Major points to be addressed:1. In Figure 1C, the co-immunoprecipitation Regnase1 with myc-tagged 14-3-3epsilon doesn't look exceptionally robust, especially when seeing the original blot provided as source data. It's currently hard to tell whether this represents a small amount of endogenous Regnase1 or a larger percentage co-precipitating. What would help to improve the strength of this figure is to include an input control on the same blots as the IPs are run, and to include a negative control blot (e.g. actin) as well. This will help to validate the specificity of the interaction.

This comment was found in the Essential Revisions and was already addressed.

2. For Figure 1D, are the authors able to show a longer exposure of the Regnase1 blots as well? For both the input blot and the blot from the immunoprecipitation. This would help to strengthen their conclusion that R848 and CpG DNA stimuli promote the interaction between 14-3-3 and Regnase1.

This comment was found in the Essential Revisions and was already addressed.

3. The authors conclude that the interaction of Regnase1 and 14-3-3 is mediated by IRAK-dependent phosphorylation. As these experiments were all conducted by overexpression of IRAK1 or 2, it would significantly strengthen their conclusion if they carried out some knockdown experiments where they deplete endogenous IRAK1/2. Does this abolish the 14-3-3 interaction with Regnase1 in IL-1B treated cells?

We thank the reviewer for the suggestion. We have previously showed that double knockout of *Irak1* and *Irak2* abolished the LPS-induced phosphorylation of Regnase-1, resulting in the disappearance of slowly migrating Regnase-1 in immunoblot analysis (Iwasaki et al., 2011). To further investigate if IRAK1/2 are necessary for the association between Regnase-1 and 14-3-3 as well as the phosphorylation of Regnase-1, we utilized CRISPR-Cas9 system by introducing pX459 plasmids, which express Cas9 and gRNAs for *IRAK1* and *IRAK2* in HeLa cells, followed by the selection with puromycin. Then we transfected HA-14-3-3ε in the puromycin-resistant cells. We confirmed that the cells lack IRAK1 expression by immunoblot analysis, although antibodies detecting endogenous human IRAK2 were not available which precluded the evaluation of *IRAK2* knockout efficiency. It is known that IL-1β stimulation induces various post translational modification of IRAK1, rendering the detection of IRAK1 unable by immunoblot analysis (Vollmer et al., 2017; PMID: 28512203). Consistently, we also did not detect the clear band of IRAK1 in IL-1β-stimulated control sample. Consistent with our previous finding using MEFs (Iwasaki et al., 2011), phosphorylation of Regnase-1 is diminished in cells transfected with pX459 for *IRAK1/2* (Figure 2—figure supplement 3). Furthermore, immunoprecipitation with HA-14-3-3ε revealed that the association of Regnase-1 and 14-3-3 was also diminished by depletion of IRAK1/2 expression. These results indicate that IRAK1/2 (at least IRAK1 in HeLa cells) is necessary for Regnase-1-14-3-3 interaction.

4. While I agree with the authors that their data support different phosphorylated serine residues supporting interactions with either 14-3-3 or TRCP, I don't think that their data specifically support mutually exclusive interactions between the two. A convincing competition experiment to test this may be to express both 14-3-3 and TRCP at different levels along with Regnase1. This would allow them to immunopreciptate a tagged Regnase 1 and see if, for example, increasing TRCP expression increases its association with Regnase1, while decreasing the association of 14-3-3 with Regnase1.

We thank the reviewer for the suggestion. The other reviewers also pointed out the wording, "mutually exclusive". As we described in our response to the Reviewer #2, we corrected the wording.

5. For Figure 5G, the authors should also test specific Serine-to-Alanine mutants to see if they impact RNA binding. This could be accomplished with a mutant that cannot bind TRCP (to keep it stable), and compare it to a mutant that cannot bind both TRCP and 14-3-3. This would allow the authors to make conclusions on a Regnase1 mutant that did not contain additional bacterial stretches to it, which could bind to proteins other than 14-3-3 that they don't know about.

We thank the reviewer for raising the concern. We agree that introduction of the ExoS sequence may cause effects on Regnase-1 other than binding with 14-3-3. The difficulty of using the S513A mutant to evaluate the functional effect of 14-3-3 on Regnase-1 is that not all the overexpressed Regnase-1 is phosphorylated by IL-1β stimulation. Particularly, IL-1β-induced phosphorylation rate of overexpressed Regnase-1 is not so high (e.g., Figure 2D, 3A, and 3B). Since RNA-immunoprecipitation requires large amount of Regnase-1 to evaluate its RNA-binding capacity, we feel it difficult to finely detect the changes in the binding of Regnase-1 WT and S513A to mRNA in response to IL-1β stimulation. Thus, utilizing an ExoS mutant and its control, an ExoSAAA mutant, was the most practical way to elucidate the mechanism how 14-3-3 affects the function of Regnase-1, although we agree that further studies are required to conclude that 14-3-3 inhibits Regnase-1-RNA binding in vivo. We discuss the requirement of future studies in the Discussion section.

6. Similarly, the final conclusion of the manuscript is that 14-3-3 binding to Regnase1 localizes to the nucleus. However, this is based on Regnase mutants that utilize bacterial stretches that bind 14-3-3 proteins (and potentially other proteins as well). To strengthen their data, the authors should utilize a Regnase mutant that cannot bind to TRCP and/or contains mutations of serine residues such that it cannot bind 14-3-3 proteins. These mutants could then be used to test if nuclear localization is impaired without adding additional bacterial protein stretches to it.

We thank the reviewer for the criticism. Similar to the aforementioned response to comment #5, the difficulty in the use of Regnase-1-overexpressing cells is that IL-1β-induced phosphorylation rate of overexpressed Regnase-1 is not so high (e.g., Figure 2D, 3A, and 3B), thus it is hard to assess the IL-1β-induced changes in intracellular dynamics of Regnase-1 by using overexpression system. Therefore, we believe that the use of the ExoS system is the most practical way to assess the role of Regnase-1-14-3-3 binding, although further studies are necessary to elucidate the function and detailed mechanism of nuclear and cytoplasmic shuttling of Regnase-1 in vivo. We discussed these issues in the Discussion section.